# Response to Hypersalinity of Four Halophytes Growing in Hydroponic Floating Systems: Prospects in the Phytomanagement of High Saline Wastewaters and Extreme Environments

**DOI:** 10.3390/plants12091737

**Published:** 2023-04-22

**Authors:** Meri Barbafieri, Francesca Bretzel, Andrea Scartazza, Daniela Di Baccio, Irene Rosellini, Martina Grifoni, Roberto Pini, Alice Clementi, Elisabetta Franchi

**Affiliations:** 1Research Institute on Terrestrial Ecosystems, National Research Council of Italy (IRET-CNR), Via Moruzzi 1, 56124 Pisa, Italy; meri.barbafieri@cnr.it (M.B.); francesca.bretzel@cnr.it (F.B.); andrea.scartazza@cnr.it (A.S.); irene.rosellini@cnr.it (I.R.); grifonimar@gmail.com (M.G.); roberto-pini@cnr.it (R.P.); 2Eni S.p.A., Subsurface and Wells R&D Projects, Via Maritano 26, San Donato Milanese, 20097 Milan, Italy; alice.clementi@eni.com; 3Eni S.p.A., R&D Environmental &Biological Laboratories, Via Maritano 26, San Donato Milanese, 20097 Milan, Italy

**Keywords:** hypersaline environment, photosynthetic pigments, fluorescence parameters, nutrient absorption, organ element distribution, sodium content, phytodesalinization

## Abstract

Hypersaline environments occur naturally worldwide in arid and semiarid regions or in artificial areas where the discharge of highly saline wastewaters, such as produced water (PW) from oil and gas industrial setups, has concentrated salt (NaCl). Halophytes can tolerate high NaCl concentrations by adopting ion extrusion and inclusion mechanisms at cell, tissue, and organ levels; however, there is still much that is not clear in the response of these plants to salinity and completely unknown issues in hypersaline conditions. Mechanisms of tolerance to saline and hypersaline conditions of four different halophytes (*Suaeda fruticosa* (L.) Forssk, *Halocnemum strobilaceum* (Pall.) M. Bieb., *Juncus maritimus* Lam. and *Phragmites australis* (Cav.) Trin. ex Steudel) were assessed by analysing growth, chlorophyll fluorescence and photosynthetic pigment parameters, nutrients, and sodium (Na) uptake and distribution in different organs. Plants were exposed to high saline (257 mM or 15 g L^−1^ NaCl) and extremely high or hypersaline (514, 856, and 1712 mM or 30, 50, and 100 g L^−1^ NaCl) salt concentrations in a hydroponic floating culture system for 28 days. The two dicotyledonous *S*. *fruticosa* and *H*. *strobilaceum* resulted in greater tolerance to hypersaline concentrations than the two monocotyledonous species *J. maritimus* and *P. australis*. Plant biomass and major cation (K, Ca, and Mg) distributions among above- and below-ground organs evidenced the osmoprotectant roles of K in the leaves of *S*. *fruticosa*, and of Ca and Mg in the leaves and stem of *H*. *strobilaceum*. In *J*. *maritimus* and *P*. *australis* the rhizome modulated the reduced uptake and translocation of nutrients and Na to shoot with increasing salinity levels. *S*. *fruticosa* and *H*. *strobilaceum* absorbed and accumulated elevated Na amounts in the aerial parts at all the NaCl doses tested, with high bioaccumulation (from 0.5 to 8.3) and translocation (1.7–16.2) factors. In the two monocotyledons, Na increased in the root and rhizome with the increasing concentration of external NaCl, dramatically reducing the growth in *J. maritimus* at both 50 and 100 g L^−1^ NaCl and compromising the survival of *P. australis* at 30 g L^−1^ NaCl and over after two weeks of treatment.

## 1. Introduction

In the last decade, increased salinization, mineral extraction, agricultural runoff, climate, urbanization, and industrial processes have expanded hypersaline environments worldwide [1,2]. Hypersaline environments are areas characterized by extreme levels of salinity where salinization has been increased by climate change or pollution. In the first case, the rising global temperatures and variable precipitations enhance the wet and dry depositions of ocean salt (NaCl) from rainfalls and aeolian flows, and they stimulate the release of salts from the physical-chemical weathering of parent rock materials. In the second case, extreme levels of salinity in hypersaline environments can be reached by poor land-water management and overuse of fertilizers, irrigation with saline-sodic and treated wastewaters, surface or subsurface sea water intrusion, and excessive discharge of treated produced water (PW) from oil and gas industrial plants [3,4]. This increased level of salinization and the decreasing availability of freshwater affect every continent, inhibiting crop productivity and dramatically reducing cultivated lands [5,6].

Soils are saline when the conductivity of saturation extract (ECe) is 4 dS m^−1^ or more, while in a hypersaline environment the salt concentration is that of full-strength sea water (>3.5–4.0%, *w/v* or 35–40 g L^−1^ NaCl) or over [2,7,8]. Sea water intrusion due to groundwater over exploitation is currently encountered in coastal areas of Italy, Spain, and Greece peninsulas, and North Africa regions [9]. Extensive coastal regions of Gulf countries (United Arab Emirates, Kuwait, Bahrain, Oman, Saudi Arabia, and Qatar) are becoming hypersaline due to intense petroleum exports and the use of open ponds for the discharge of treated PW [10]. Indeed, apart from the presence of hydrocarbon residues, one of the main issues in the PW management is the extremely high salinity, with an average range of 1–250 g L^−1^, that is, from 17 to 4278 mM NaCl [11].

The extremely high levels of NaCl reached in hypersaline environments are unfavorable for the growth of most organisms, including microorganisms, and thus limiting sustainable options such as bio- and phytoremediation for the reclamation of such areas or treatment of PW [4,11,12]. The only plants able to grow in saline conditions are the halophytes, which constitute 1% of the world flora and are among the most widely used species in phytodesalinization or salt phytoremediation, as not only can they tolerate high levels of salt but they are also able to complete their life cycle in a salt concentration of at least 200 mM NaCl and to achieve optimal growth levels at NaCl concentrations between 200 and 1000 mM (i.e., 12 and 58 g L^−1^, respectively) [13,14]. Such NaCl concentrations are high and close to those of saline (5–35 g L^−1^ NaCl) soil and water conditions of many world areas and to the increasing hypersaline (>35 g L^−1^ NaCl) areas described above. Tolerance responses of halophytes to NaCl stress depend on several interacting variables, including plant species, plant developmental stage, NaCl concentration, time of exposure, and growth environment. In this context, many studies have been conducted on salinity tolerance in halophytes and their mechanisms to counteract salt stress (e.g., [13,15,16] and literature within). However, these works rarely investigate halophytes response to salt concentrations over 600 Mm (or 35 g L^−1^) NaCl, corresponding to seawater in hydroponic culture conditions [17]. Some other works are germination tests or studies on the geographical distribution of halophytes in areas where salinity can be extremely high (coasts, salt marshes) but not constant, e.g., [18,19,20,21]. Thus, major aspects of the halophytes’ response to hypersalinity remain unknown. Conversely, some halophytes have been tested in the phytoremediation of saline waters and soils contaminated by organic pollutants, as in the case of *Juncus maritimus* or *Phragmites australis* for PW containing petroleum-hydrocarbon residues [22,23], *Halocnemum strobilaceum* in oil-polluted contaminated sites [24], or *J*. *roemerianus* and *J*. *acutus* for diesel fuel in saline soil [25,26]. Other studies show that halophytes have already been successfully used for the remediation of heavy metal-polluted soils affected by salinity [8,27]. However, before evaluating the effectiveness of halophytes in terms of remediation of such contaminants, it is necessary to assess the plants’ ability to survive and tolerate the specific saline conditions of soils and (waste)waters.

Halophytes can adapt to saline environments by different mechanisms at the cell, tissue, and organ levels, and recent genetic and molecular research has shown complicated regulatory networks coordinating halophyte tolerance and adaptation to salt stress [16,28]. The first effects of salinity on plants are stomatal closure and photosynthesis inhibition, with consequent growth reduction, dehydration, nutritional imbalance, selective ion uptake and distribution, including sodium (Na) [7,13]. A screening of the hypersaline response in halophytes is expected to firstly investigate such physiological and chemical characteristics. Emerging sequencing techniques and platforms will help to better characterize all components of such responses in detail and eventually transfer these functions to more sensitive plants; however, genetic manipulation of several genes to enhance the required trait can be time-consuming and less successful than physiological selection.

In this study, we tested the tolerance to high saline and hypersaline conditions of four Mediterranean halophytes, two dicotyledonous, *Suaeda fruticosa* L. (Forssk) and *Halocnemum strobilaceum* (Pall.) M. Bieb., and two monocotyledonous rhizomatous species, *Juncus maritimus* Lam. and *Phragmites australis* (Cav.) Trin. ex Steudel. The plants were grown under hydroponic floating conditions, in the presence of high NaCl levels, simulating the concentrations of marine and/or waste waters characterized by high salinity, such as PW. The NaCl stress tolerance was assessed by monitoring growth parameters (biomass production and partitioning), physiological and biochemical traits (photochemical efficiency of photosystem II, photosynthetic pigments), and the distribution of main nutrients, cations, and sodium. The main aim of this work was to screen the salt tolerant capacity to hypersaline conditions of these four halophytic species and Na accumulation in different organs. Specific goals of this research were to identify: (i) the more tolerant halophytic species to hypersaline stress among four native Mediterranean species; (ii) the NaCl salinity threshold at which such species maintain their viability; and (iii) the Na accumulation capacity of halophytes exposed to hypersaline conditions for phytodesalinization perspectives. Several authors have studied halophytes tolerance to salt stress; however, there are few reports on such plant growth, photosynthetic traits, and element distribution in different organs under hypersaline treatments, and, as far as we know, no comparative studies including these different four species. In line with the concepts expressed above, the results of this study will contribute to select halophytic plant species tolerant to extremely high amounts of salt and thus suitable for the phytomanagement of hypersaline areas and high-saline industrial wastewaters.

## 2. Results

### 2.1. Growth, Biomass Partitioning, and Water Content

Botanical and ecological characteristics of the four halophytic species used in the experiment are reported in Table 1. During 28 days of NaCl treatment in the hydroponic system, signs of wilting and chlorosis appeared in the shoots of *P*. *australis* (subjected to 30, 50 and 100 g L^−1^ NaCl) and *J*. *maritimus* (50, and 100 g L^−1^ NaCl). In particular, the shoot of *P*. *australis* under the higher NaCl concentrations of 50 and 100 g L^−1^ was severely impaired after 14 days of growth. Minor visible damage signs (e.g., sporadic chlorotic spots) were observed in the leaves of *S*. *fruticosa* and *H*. *strobilaceum* under 100 g L^−1^ NaCl. The NaCl levels of 0, 15, 30, 50, and 100 g L^−1^ used as treatments correspond to 0, 257, 214, 856, and 1712 mM NaCl, respectively. For the indication of these treatments throughout the work, the expression in mg of salt per litre was chosen because it is mostly used in the description of the salinity in hypersaline environments and highly saline wastewater [4,11]. However, in tables and graphs, the correspondence between the expression on weight and on a molar basis is always indicated.

At harvest (Appendix A), the total biomass production and partitioning among above- (leaf and stem or shoot) and below-ground (rhizome and/or root) organs were simultaneously influenced by differences between NaCl concentrations and plant species factors and their interactions (Figure 1). In *S*. *fruticosa*, the foliar DW was reduced by 63, 62, and 75.5% when treated with 30, 50, and 100 g L^−1^ NaCl, respectively (Figure 1A). In *H*. *strobilaceum*, leaves decreased by 16% at 50 g L^−1^ NaCl and 57% at 100 g L^−1^ NaCl treatment; in *J. maritimus*, the shoot was reduced starting from 30 g L^−1^ NaCl treatment (−35%) and continued under 50 and 100 g L^−1^ NaCl (about −63% in both cases). In *P*. *australis*, the shoot was dramatically affected by 30, 50, and 100 g L^−1^ NaCl treatments, with 70.5, 54, and 67% decreases, respectively (Figure 1A).

The stem of *S*. *fruticosa* was reduced (−38%) only at 100 g L^−1^ NaCl, while in *H*. *strobilaceum* it was not affected (Figure 1B). The rhizome biomass of *J*. *maritimus* remained unaltered after NaCl applications, and in *P*. *australis* it was reduced by 45% under 50 g L^−1^ NaCl and by about 57% at both 30 and 100 g L^−1^ NaCl treatments (Figure 1B). The roots of *S*. *fruticosa* decreased by 46% under 30 g L^−1^ NaCl and by 30–31% under both 50 and 100 g L^−1^ NaCl; in *H*. *strobilaceum*, the roots were affected (−52%) only at 100 g L^−1^ NaCl treatment (Figure 1C). The root biomass of *J*. *maritimus* was reduced at 50 and 100 g L^−1^ NaCl (−48 and −40%, respectively), while in *P*. *australis* it was not affected by NaCl treatments (Figure 1C).

In *S*. *fruticosa*, the total biomass was reduced from 30 g L^−1^ NaCl, with the highest extent (−49%) at 100 g L^−1^ NaCl, while in *H*. *strobilaceum* it was affected only at 100 g L^−1^ NaCl (−44.4%, Figure 1D). The total biomass of *J*. *maritimus* decreased by 43% at 50 g L^−1^ NaCl and by 35.5% at 100 g L^−1^ NaCl; in *P*. *australis*, the biomass was reduced by 57, 43.4, and 55% in the presence of 30, 50, and 100 g L^−1^ NaCl, respectively (Figure 1D). Further parameters on the relative water status and biomass partitioning of the four halophytes exposed to high saline and hypersaline NaCl concentrations compared to controls are shown in the Appendix A. The main results were as follows:

The relative water content (RWC) was mainly dependent on differences among the plant species (Appendix A). The relative biomass partitioning between the aerial parts and roots of the four halophytes was influenced by differences between NaCl treatments and plant species and their interaction (Appendix A). In *S*. *fruticosa*, the root-to-shoot ratio increased (54.4%) only at 100 g L^−1^ NaCl treatment. In both *J*. *maritimus* and *P*. *australis*, the root/shoot ratio increased starting from a 30 g L^−1^ NaCl treatment (Appendix A). In *S*. *fruticosa*, *H*. *strobilaceum*, and *J*. *maritimus*, the shoot mass ratio (SMR) reflected the trend of the root to shoot ratio, while in *P*. *australis* it was relatively low and unchanged compared to the control (Appendix A).

### 2.2. Chlorophyll Fluorescence and Photosynthetic Pigments

Fluorescence parameters are reported in Figure 2 and Figure 3. The Fv/Fm in *H*. *strobilaceum* was not significantly different from the control plants (close to 0.8), up to 50 g L^−1^ NaCl concentration throughout the treatment period, while at 100 g L^−1^ it showed a progressive decrease starting from the first week (0.634) to a minimum after 21 days (0.113) of treatment, remaining unchanged in the last week (Figure 2). In *S*. *fruticosa*, Fv/Fm showed a similar behavior as in *H*. *strobilaceum*, but at 50 g L^−1^ NaCl was only slightly lower than control starting from the third week, and at 100 g L^−1^ NaCl sharply declined in the second week (0.247) and continuously decreased until the last week (0.112). In *J*. *maritimus*, Fv/Fm did not change up to 30 g L^−1^ NaCl, while at higher NaCl concentrations, decreased from the first week to the fourth, reaching values close to 0. *P*. *australis* maintained Fv/Fm values similar to control throughout the treatment period only at 15 g L^−1^ NaCl, while Fv/Fm progressively decreased to values close to 0 with the increasing salt concentration and treatment time. In *H*. *strobilaceum*, ΦPSII values in plants treated with NaCl concentrations higher than 15 g L^−1^ NaCl progressively decreased, reaching at the end of the treatment period values of 0.582, 0.464, and 0.215 for 30 g L^−1^, 50 g L^−1^, and 100 g L^−1^ NaCl, respectively (Figure 3). In *S*. *fruticosa* treated with 15 and 30 g L^−1^ NaCl, the ΦPSII did not change when compared to control (around 0.700), while at 50 and 100 g L^−1^ NaCl, it decreased to 0.435 and 0.065, respectively, at the end of treatment. In *J*. *maritimus* treated with 50 g L^−1^ and 100 g L^−1^ NaCl, the ΦPSII sharply decreased between the first and the second week, reaching values close to 0. In *P*. *australis*, the ΦPSII was unchanged only at 15 g L^−1^ NaCl, while at higher NaCl concentrations it dropped, reaching values close to 0 in the first week (100 g L^−1^ NaCl) and the last week (30 and 50 g L^−1^, Figure 3).

The trends of ΦPSII reflected those of the NPQ but in the opposite direction (Figure 3E–H). In *H*. *strobilaceum* and *S*. *fruticosa*, NPQ reached a peak at 100 g L^−1^ NaCl after 7 (~1.2) and 14 (~1.6) days of treatment, respectively, followed by a sharp decrease with the prolonged stress duration. Conversely, at 50 g L^−1^ NaCl, NPQ increased starting from the second week and remained higher than the other treatments until the last week. In *J*. *maritimus*, both 50 and 100 g L^−1^ NaCl treatments caused a peak of the NPQ after 7 days (~0.8) followed by a sharp decrease, while at lower NaCl concentrations (30 and 15 g L^−1^ NaCl), the NPQ reached a peak after 21 days. In *P*. *australis*, NPQ decreased just after 7 days of salt treatments above 15 g L^−1^ NaCl.

Changes in photosynthetic pigments in the four halophytic species in response to increasing NaCl concentrations were investigated (Figure 4 and Figure 5; Appendix A). In *S*. *fruticosa*, the leaf total chlorophyll (Chl *a* + *b*) content was reduced in the presence of NaCl at 7 days of treatment (Figure 4A), and at the same extent (about −42%) with 15, 30, and 50 g L^−1^ NaCl and by 85% with 100 g L^−1^ NaCl. In the following weeks, the Chl *a* + *b* reduction continued to be remarkably evident from 30 to 100 g L^−1^ NaCl (Figure 4A). These decreases were due to both Chl *a* and Chl *b* reductions (Appendix A). The Chl *a* to *b* ratio was altered in *S*. *fruticosa* leaves between 50 and 100 g L^−1^ NaCl from the second week of treatment to the end of exposure (Figure 4E). In *H*. *strobilaceum* leaves, the Chl *a* + *b* was affected by a general reduction in all the NaCl doses during the first week of treatment (Figure 4B). Then, the leaf Chl *a* + *b* of *H*. *strobilaceum* under 15 g L^−1^ NaCl progressively increased, reaching the same levels of control from 14 to 28 days. In the same species, Chl *a* + *b* contents became unaltered compared to controls also in the presence of 30 g L^−1^ NaCl during the last week of NaCl exposure, while they remarkably decreased at 50 and 100 g L^−1^ NaCl with the higher extent (−74%) in the presence of 100 g L^−1^ (Figure 4B). Correspondingly, the Chl *a*/*b* ratio was affected only with 50 and 100 g L^−1^ NaCl at the end of treatment (Figure 4F). In *J*. *maritimus* leaves, after the first week of treatment, in which Chl *a* + *b* decreased only in the presence of 50 and 100 g L^−1^ NaCl (−22% in both cases), Chl *a* and Chl b showed different rates of decrease at all NaCl doses tested until the end of exposure (Figure 4C; Appendix A), reaching a decline between 15 and 30 g L^−1^ NaCl (about −63%), and between 100 g L^−1^ and 50 g L^−1^ NaCl (−95 and −86.5%, respectively). In the same species, the Chl *a*/*b* was affected by NaCl at different time points of treatment, with the highest decreases at 50 and 100 g L^−1^ NaCl (−52 and −54%, respectively) during the last week (Figure 4G). In *P*. *australis*, the Chl *a* + *b* was maintained at a relatively low level (under 5 mg g^−1^) indifferently at all the NaCl doses during the first two weeks of exposure (Figure 4D). Then, *P*. *australis* exposed to 30 g L^−1^ NaCl and superior doses was affected by a dramatic decrease in growth and photosynthetic performance; consequently, on the leaves of such plants, it was not possible to determine the concentration of photosynthetic pigments (Figure 4 and Figure 5; Appendix A). The shoot Chl *a* + *b* of *P*. *australis*, subjected to 15 g L^−1^ NaCl, maintained control levels until 21 days of NaCl treatment, while it decreased (−16%) at the end of the experiment (Figure 4D), due to a contemporary decrease of Chl *a* and Chl *b* (Appendix A). In this species, the Chl *a*/*b* ratio was differently impaired since the first week of NaCl treatment at all the doses tested, with a slight decrease or unchangeable levels at 15 g L^−1^ NaCl (Figure 4H).

In *S*. *fruticosa*, the leaf total carotenoid content was reduced (−81%) during the first week of treatment to only under 100 g L^−1^ NaCl (Figure 5A). In the same species, after one week of apparent recovery, carotenoids started to decrease under the effects of NaCl treatments, with the only exception of 15 g L^−1^ NaCl dose, at which these pigments increased, reaching the control levels at 28 days (Figure 5A). In *H*. *strobilaceum*, after a decrease during the first week of treatment under 15, 50, and 100 g L^−1^ NaCl, the leaf carotenoid content was completely recovered after 14 days of treatment, followed by a new decrease at all the NaCl doses (14–21 days); in the last week, carotenoids went on decreasing at 50 and 100 g L^−1^ NaCl (Figure 5B). In the shoot of *J*. *maritimus*, carotenoids were influenced by NaCl treatments at all the time points investigated (Figure 5C). They first increased (0–7 days) in the presence of 50 and 100 g L^−1^ NaCl (38 and 20%, respectively), and then (7–14 days) only in the presence of 30 g L^−1^ NaCl (21%). Subsequently, the shoot carotenoids of *J*. *maritimus* started to decrease at all the NaCl doses tested until the end of treatment (Figure 5C). In *P*. *australis*, the shoot carotenoids were not affected by NaCl treatments during the first two weeks of exposure (Figure 5D). Then, as for chlorophylls, the physiological and visible damage symptoms shown by *P*. *australis* in the two last weeks of NaCl treatment did not permit the determination of carotenoids in the shoots of plants treated with 30, 50, and 100 g L^−1^ NaCl (Figure 5D). In presence of 15 g L^−1^ NaCl, the carotenoid contents in *P*. *australis* shoots showed the same trend found for chlorophylls (Figure 4D and Figure 5D).

### 2.3. Nutrients and Sodium Contents: Plant Organ Distribution and Accumulation

In the aerial parts, *S*. *fruticosa* showed the highest N concentration, followed by *P*. *australis*, *H*. *strobilaceum*, and *J*. *maritimus* (Table 2). In all the halophytes, the leaf or shoot N concentration decreased with the increase in NaCl dose, with elevated extents in the presence of 30 g L^−1^ NaCl in *S*. *fruticosa* (−32%), *H*. *strobilaceum*, and *J*. *maritimus* (−24% in both of them), and at 50 g L^−1^ NaCl in *P*. *australis* (−37%). The stems of *S*. *fruticosa* and *H*. *strobilaceum* contained, on average, the highest N levels (1.84 and 1.45%), independently of the NaCl concentration. In both the rhizomatous species, the N rhizome concentration decreased between 50 and 100 g L^−1^ NaCl. Control plants of *P*. *australis* showed the highest N content in the roots (3.59%), but it was dramatically reduced starting from 15 to 100 g L^−1^ NaCl treatment (−40–70%). In *J*. *maritimus*, the N root concentration started to decrease after 30 g L^−1^ NaCl, in *H*. *strobilaceum* after 50 g L^−1^, and in *S*. *fruticosa* only at 100 g L^−1^ (Table 2).

In both *S*. *fruticosa* and *H*. *strobilaceum*, the P concentration increased in all the analysed organs at 100 g L^−1^ NaCl, and in the stem in presence of 50 g L^−1^ NaCl; in the root of *S*. *fruticosa*, P enhanced from 15 to 100 g L^−1^ NaCl (Table 2). In the shoot and rhizome of *J*. *maritimus*, P concentration tended to increase, with the exception of the shoot at 15 g L^−1^; in the roots, P was very low (on average, 0.22 mmol 100 g^−1^ DW) or below the detection level. In *P*. *australis* shoots, the P concentration was reduced (−13.4%) in presence of 50 g L^−1^ NaCl and in both rhizome and root at 100 g L^−1^ NaCl (−35 and −70%, respectively, Table 2).

The leaf K concentration of *S*. *fruticosa* increased with the increasing of NaCl treatment, while it tended to decrease in the stem and root, with the only exception of root in presence of 15 g L^−1^ NaCl (Table 2). In *H*. *strobilaceum*, K increased in both shoot and stem at 50 g L^−1^ NaCl and decreased at 100 g L^−1^ NaCl; in the root, K progressively decreased with the increasing NaCl treatment. In *J*. *maritimus* shoot, K concentration increased (25–30%) starting from 30 g L^−1^ NaCl treatment, and in the rhizome, it enhanced at 50 and 100 g L^−1^ NaCl (2–2.5-fold higher, Table 2). In *P*. *australis* shoot and root, K progressively decreased from 30 to 100 g L^−1^ NaCl; in the rhizome, K tended to increase with the increasing NaCl treatment, reaching about 11.0 mmol 100 g^−1^ DW at 100 g L^−1^ NaCl (Table 2).

*S*. *fruticosa* and *H*. *strobilaceum* contained the highest amounts of Ca and Mg in leaves and roots (Figure 6A,C,D,F). In the leaves of *S*. *fruticosa*, both Ca and Mg decreased after 15 g L^−1^ NaCl treatment, and then started to increase until the highest NaCl dose of 100 g L^−1^ NaCl: Ca exceeded control levels, while Mg remained below (Figure 7A,D). In *H*. *strobilaceum*, the leaf Ca concentration progressively increased with the increasing NaCl treatment, while Mg was reduced (−11.5%) at 30 g L^−1^ NaCl and enhanced (36%) at 100 g L^−1^ NaCl (Figure 6D). In *J*. *maritimus* treated with 100 g L^−1^ NaCl, shoot Ca concentrations increased (38%) while they slightly decreased (−16%) in *P*. *australis* under 50 g L^−1^ NaCl (Figure 6A). In *J*. *maritimus* shoots, Mg increased in the presence of 30 and 50 g L^−1^ NaCl and in *P*. *australis* it progressively decreased from 15 to 100 g L^−1^ NaCl (Figure 6D). In the stem of *S*. *fruticosa*, Ca gradually increased with NaCl treatment, while Mg decreased (about −22%) at both 30 and 50 g L^−1^ NaCl and increased (50%) at 100 g L^−1^ NaCl (Figure 6B,E). In the stem of *H*. *strobilaceum*, both Ca and Mg concentrations increased (19 and 37%, respectively) at 100 g L^−1^ NaCl. In the rhizome of *J*. *maritimus*, Ca concentration was affected by NaCl treatments with an average reduction range of 30–48% compared to control; in the same organ, Mg was reduced by 40–47% in the presence of 15, 30, and 100 g L^−1^ NaCl (Figure 6B,E). In *S*. *fruticosa* roots, both Ca and Mg concentrations were reduced by NaCl treatments, with the highest extents at 100 g L^−1^ NaCl (−71 and −76% for Ca and Mg, respectively, Figure 6C,F). In *H*. *strobilaceum* roots, Ca increased (19%) at 30 g L^−1^ NaCl and decreased (about −31%) in the presence of 50 and 100 g L^−1^ NaCl (Figure 6C); in the same organ, Mg was enhanced at 50 g L^−1^ NaCl and reduced at 100 g L^−1^ NaCl to the same extent (about 20%, Figure 6F). In *P*. *australis* roots, Ca concentration increased (42%) in the presence of 30 g L^−1^ NaCl and drastically decreased (−71%) at 100 g L^−1^ NaCl (Figure 6C), while Mg was reduced by 69% under 100 g L^−1^ NaCl treatment (Figure 6F).

The highest K/Na values were found in the shoot of the two rhizomatous species, in the stem of *S*. *fruticosa*, in the rhizome of *P*. *australis*, and in the root of *S*. *fruticosa*, *H*. *strobilaceum*, and *P*. *australis* (Table 3). This ratio was progressively enhanced (from 0.11 to 1.80) by the increasing NaCl concentration only in the leaves of *S*. *fruticosa*. The Ca/Na ratio was particularly elevated in the roots of *H*. *strobilaceum* (about 7) and *J*. *maritimus* (7.5) and relatively high in the rhizome and root (average of 3.6) of *P*. *australis* (Table 4). In both *S*. *fruticosa* and *H*. *strobilaceum* leaves, the Ca/Na ratio increased with the enhancement of the NaCl dose, while decreasing in the stem and root (Table 3). In all the organs of the two rhizomatous species, the Ca/Na ratio decreased with the increasing NaCl concentration, with the highest rate in the root (Table 3).

In the aerial parts, the highest Na concentrations (mmol Na 100 g^−1^ DW) were found in the leaves of *H*. *strobilaceum* (control and treated plants) and *S*. *fruticosa* (control and treated with 15, 30, and 50 mg L^−1^ NaCl), in the shoots of *J*. *maritimus* and *P*. *australis* exposed to 50 and 100 g L^−1^ NaCl, respectively (Figure 7A). In the stem of *S*. *fruticosa*, the Na concentration progressively increased with the enhancement of the NaCl dose, while in *H*. *strobilaceum* it started to increase from 30 to 100 g L^−1^ NaCl (Figure 7B). In both *J*. *maritimus* and *P*. *australis* rhizomes, Na concentration gradually increased with the increasing NaCl treatment (Figure 7B). In the root of all four halophytes, Na concentration progressively enhanced with the increase of NaCl dose, reaching in any case the highest levels in the presence of 100 g L^−1^ NaCl (Figure 7C).

The leaf Na content or removal (mg Na organ or plant^−1^) in *S*. *fruticosa*, after an increase of 15 g L^−1^ NaCl (13%), decreased with the increase in NaCl dose (Figure 7D). In the stem of the same species, Na content almost quadrupled at 15 g L^−1^ NaCl, and it maintained this level until 100 g L^−1^ NaCl treatment (Figure 7E); in the root of *S*. *fruticosa*, Na uptake was progressively enhanced with the increasing NaCl concentration (Figure 7F). In *H*. *strobilaceum* leaves, Na content decreased by 16 and 58% in the presence of 50 and 100 g L^−1^ NaCl, respectively (Figure 7D), while in the stem and root it increased at 100 g L^−1^ NaCl (Figure 7E,F). In *J*. *maritimus*, the shoot Na content increased at 15, 30, and 50 g L^−1^ NaCl, and it gradually enhanced in the rhizome and root with the increasing NaCl treatment (Figure 7D–F). In *P*. *australis*, Na accumulated in the rhizome of plants treated with 15, 50, and 100 g L^−1^ and it increased in the root at 50 and 100 g L^−1^ NaCl (Figure 7D–F). The total Na removal in *S*. *fruticosa* and *H*. *strobilaceum* was generally high, with the highest extent at 15 g L^−1^ NaCl for the former (206.6 mg Na plant^−1^) and at 30 g L^−1^ NaCl for the latter (117.5 mg Na plant^−1^, Figure 7G). In the two rhizomatous species, Na accumulation progressively increased with the enhancement of NaCl treatment, with relatively higher levels in *J. maritimus* in the presence of 30, 50 and 100 g L^−1^ NaCl (Figure 7G).

The highest values for bioaccumulation (BAF) and translocation (TF) factors of Na were found in *S*. *fruticosa* and *H*. *strobilaceum* (Table 4). In both species, the BAF determined in the shoot (BAF_shoot_) was over 1, except in the presence of 100 g L^−1^ NaCl. In *S*. *fruticosa* and *H*. *strobilaceum*, the BAF in the root (BAF_root_) was relatively low (0.11–0.87).

The total BAF (BAF_whole plant_) was under 1 only in *H*. *strobilaceum* exposed to 100 g L^−1^ NaCl. In both *S*. *fruticosa* and *H*. *strobilaceum*, the TF was higher than 1 in any of the conditions analysed (Table 4). In *J*. *maritimus*, the BAF_whole plant_ was over 1 at all NaCl treatments due to the contribution of shoots and underground organs, while in *P*. *australis* treated plants it was always under 1. The Na TF was always under 1 in both the rhizomatous species, and NaCl treatments progressively reduced the BAF and TF in all their organs (Table 4).

The results for the correlations between the main physiological and biochemical attributes determined in the aerial parts of the four halophytes in response to NaCl treatments are shown in Appendix A.

## 3. Discussion

### 3.1. Effects of Hypersalinity on Growth and Biomass Partitioning

The definition and classification of halophytes involve their growth and survival capacities under saline conditions, in natural or experimentally controlled environments [13,15,16]. Here, NaCl levels for saline conditions (saline-sodic soils with EC > 4 dS m^−1^, and the NaCl stress threshold for plants is 80 mM or ~5 g L^−1^) are over the limit, and the territory of hypersaline environments (over ~35 g L^−1^ NaCl, that is, ~600 mM NaCl) is reached [13,32]. The NaCl threshold for halophytes’ salinity tolerance and optimal ecophysiological growth condition is in the range of 100 to 200 mM (i.e., 6–12 g L^−1^) NaCl, but differences exist among species, ecotypes, developmental stages, and environments, so that many aspects of their underlying mechanisms remain unclear or unknown, especially under hypersaline conditions [13,19,33]. Many dicotyledonous halophytes show optimal growth in the presence of 50–250 mM NaCl, while monocotyledonous halophytes generally grow optimally in the absence of salt or at low (50 mM or less) NaCl concentrations [13,34]. Here, *S*. *fruticosa* maintained the same total DW and biomass partitioning as the control under saline conditions (15 g L^−1^ or 257 mM NaCl), showing a relevant decrease of the total biomass only after 30 g L^−1^ NaCl (i.e., 513.3 mM) exposure. In any case, the relative water content (RWC) was unaltered in all organs of *S*. *fruticosa* in respect to NaCl treatments. These results agree with previous studies [35,36], demonstrating that moderate concentrations of NaCl can stimulate the growth of *S*. *fruticosa*, which is instead more sensitive to higher NaCl concentrations. Some authors [35], for example, found that the fresh weight of roots and the fresh and dry weight of the shoots of *S*. *fruticosa* increased by saline conditions (200 to 400 mM) and decreased by high salt concentrations (600, 800, and 1000 mM). In experiments by [36], the biomass production of *S*. *fruticosa* grown hydroponically under saline conditions was significantly enhanced by salt concentrations of 100 and 400 mM. *Halocnemum strobilaceum* was impaired only at the highest NaCl treatment (100 g L^−1^ or 1712 mM NaCl), when an appreciable reduction was observed in the leaf mass but not in the stem or root; however, the relative biomass partitioning among leaves, stem, and roots was not altered by NaCl doses in comparison with controls, as well as the RWC of all the organs analysed. Several studies [37,38,39,40] have been conducted on the germination of *H*. *strobilaceum* seeds under saline conditions, whereas few works focus on this species response at other phenological stages [38,41]. However, interesting works on relationships between the composition and distribution of vegetation and environmental salinity identify *H*. *strobilaceum* among the most tolerant species to soil salinity and establish its relevance as a useful indicator for monitoring activities [18,20,42]. Both *S*. *fruticosa* and *H*. *strobilaceum* are euhalophytes, that is, obligate halophytes, and the former is a succulent type while the latter is a xerophytic one with partially succulent shoots [41,43]. These characteristics (leaf succulence in *S*. *fruticosa* and xerophytic water regime in *H*. *strobilaceum*) are consistent with the relative lower biomass reduction and dehydration of the two dicotyledonous plants tested.

The monocotyledonous *J*. *maritimus* and *P*. *australis* are two halophytes and rhizomatous perennial plant species that can grow in muddy and subaquatic saline, brackish, and freshwater conditions [43,44]. Here, *J*. *maritimus* was among the species that produced the highest total biomass up to 30 g L^−1^ NaCl and was affected by dramatic decreases between 50 and 100 g L^−1^ NaCl, for reductions in shoot and root biomass while keeping the rhizome unaltered. In other experiments, *J*. *maritimus* plants reduced their growth in the presence of 100 mM (5.8 g L^−1^) NaCl, survived at high NaCl concentrations (400–500 mM NaCl), but optimal growth was registered in the absence of salt [45]. Under the same conditions, the lower salinity levels of 75 mM (4.4 g L^−1^) NaCl stimulated the growth of *J*. *maritimus*, while it was inhibited at higher NaCl concentrations (150 and 300 mM NaCl) [46]. *Juncus maritimus* is able to grow in salt marshes and wetlands in the temperate regions of the world, including the Mediterranean Basin [31,42]. However, in both lab-scale and field work, *J*. *maritimus* grew under moderate or high salinity levels, never in hypersaline conditions (>30–35 g L^−1^, i.e., 513–600 mM) comparable to those studied in our experiment. To our knowledge, only a few reports have set forth results on one *Juncus* species, *J*. *roemerianus* Scheele, subjected to hypersalinity; such species are distributed in habitats naturally characterized by tidal flushing and seawater intrusions [47,48]. In this respect, some authors demonstrated that the salt resistance of this rush species to maritime marshes relied on its perennial life form and ability in vegetative reproduction by vigorous rhizome growth, and thus leading to the formation of dense stands [47]. This finding is consistent with our results on *J*. *maritimus*, in which the growth of the rhizome was not reduced even at the highest hypersaline treatments, compensating for the strong reduction of shoots and roots in the production of total plant biomass that remained unchanged, up to 30 g L^−1^ NaCl.

In previous works, authors [49] found that NaCl treatments of 5, 10, and 20 g L^−1^ NaCl for 100 days significantly decreased the growth of *P*. *australis*; other authors [50] showed a reduction of biomass allocation in the leaves, stems, and roots of *P*. *australis* subjected to 300 and 500 mM (i.e., 17.5–29.2 g L^−1^) NaCl for 21 days. In a screening for salt tolerance of eight halophytes, the salt-injury rate in seedlings of *P*. *australis* increased from 4.5% at 150 mM NaCl to almost 100% at 300 mM NaCl [51]. *P*. *australis* survived in presence of 12 g L^−1^ (i.e., 205 mM) NaCl in a vertical-flow constructed wetland system for the treatment of saline wastewater from the petroleum industry [23]. Although a direct comparison of the results cannot be made, since the experimental tests were performed under different conditions and at different plant phenological stages, the growth of *P*. *australis* seems to be sensitive to moderate and high NaCl concentrations ranging from 300 to 500 mM and over, as observed in our study testing high (15 g L^−1^ NaCl) and hypersaline (30–100 g L^−1^ NaCl) conditions.

### 3.2. Salt-Induced Modifications in Photosynthetic Characteristics

Overall, the Fv/Fm results highlighted a different capacity of the species studied to counteract salt stress, avoiding photoinhibition and photodamage to PSII [52], with *S*. *fruticosa* and *H*. *strobilaceum* showing the highest tolerance of the photosynthetic apparatus to hypersaline conditions. Although the stress-related decline in PSII photochemistry with consequent PSII photoinhibition occurred at 100 g L^−1^ NaCl, these two species displayed mechanisms able to prevent PSII photodamage and dissipate the excess energy at lower NaCl concentrations. The values of ΦPSII, giving an estimate of electron transport rate at PSII level [53], changed proportionally to the rate and duration of the stress applied, with a progressive decline from 15 g L^−1^ to 100 g L^−1^ NaCl, similar to what was observed for Fv/Fm, although with differences among species. The decreasing trend of ΦPSII in the four halophytic species was associated with an increase of the NPQ. Indeed, to avoid possible damages to the photosynthetic apparatus, the excess excitation energy at PSII under stress conditions must be safely dissipated by plants through a non-radiative and harmless pathway as heat [54,55,56], which is estimated by means of the NPQ. This process has been found an effective mechanism to avoid ROS (reactive oxygen species) formation, protecting the photosynthetic apparatus under high salinity conditions [57]. Moreover, it has been demonstrated that chloroplasts of halophytes, such as the euhalophytic species *Haloxylon ammodendron* and *Suaeda physophora*, are protected under high salinity conditions due to strong ion compartmentalization; thus, the ultrastructure of thylakoids in the chloroplasts of such species is not affected under stress-salt conditions [58]. However, our data suggest that with increasing salt levels and treatment time, the NPQ mechanism was not able to counteract the excess excitation energy at PSII; hence, plants were subjected to chronic photoinhibition and irreversible photodamage depending on the species and stress duration. Indeed, *J*. *maritimus* failed to dissipate the excess energy through NPQ after two weeks of treatment at both 50 and 100 g L^−1^ NaCl, while in *P*. *australis* NPQ and Fv/Fm decreased just after the first week of treatment, starting from 30 g L^−1^ NaCl. Conversely, *H*. *strobilaceum* and *S*. *fruticosa* showed a decrease in NPQ only at the highest NaCl concentration of 100 g L^−1^ starting from the first and the second weeks of treatment, respectively. Our data clearly demonstrated that *H*. *strobilaceum* and *S*. *fruticosa* showed a higher ability to dissipate the excitation energy under salt stress conditions through both radiative and non-radiative light energy dissipation processes than *J*. *maritimus* and *P*. *australis*, hence preserving the photosynthetic functionality at high NaCl concentrations for a longer period.

Salinity impairs biosynthesis and/or provokes the accelerated degradation of photosynthetic pigments, leading to photo-inhibition [59]; therefore, one of the most general effects of salt stress reported for halophytic grasses is the decrease of chlorophyll and carotenoid contents [19,60]. Indeed, in all the halophytes tested here, the total chlorophyll (Chl *a* + *b*) was affected by the higher NaCl treatments, although with different patterns and extents, depending on the species, NaCl concentration, and time of exposure. In *P*. *australis*, Chl *a* + *b* did not change during the first two weeks of NaCl treatment, but later the shoots were so damaged in the presence of 30, 50, and 100 g L^−1^ NaCl that it was not convenient to collect samples for pigment determinations by subtracting photosynthesizing surface to plants. As expected, the Chl *a* + *b* content in *S*. *fruticosa* and *H*. *strobilaceum* was generally more elevated than in the two rhizomatous monocots [41,61,62], this feature being particularly evident in *S*. *fruticosa*. In addition, the relative decreases of Chl *a* + *b* were lower and more gradual with increasing NaCl treatments in *S*. *fruticosa* and *H*. *strobilaceum* than in *J. maritimus* and *P. australis*. The functionality and viability of photosynthetic apparatus depend on the pigment composition of the two photosystems (PSs) and their light-harvesting complexes (LHCs), that is, their relative contents of Chl *a*, Chl *b* and carotenoids [56,63]. In this respect, the chlorophyll concentration and the relative Chl *a* and Chl *b* components of the photosynthetic apparatus (Chl *a*/*b* ratio) are used to quantify premature leaf senescence and changes in the leaf properties of salt-stressed plants [64,65]. In our experiment, the Chl *a*/*b* ratio in *S*. *fruticosa* decreased only in the presence of the higher NaCl concentrations investigated (50 and 100 g L^−1^ NaCl) after two weeks of treatment. In *H*. *strobilaceum*, it was significantly reduced only in the presence of 100 g L^−1^ NaCl at the end of treatments while maintaining stablity in all the other conditions. This is in agreement with the high salt tolerance of the two dicot halophytes, since the greater Chl *a*/*b* ratio has been reported as the salt tolerance index in plants grown under high soil salinity as compared to that in lower saline conditions [64,66], and the Chl *a*/*b* stability is a sign of efficient distribution of the two chlorophylls between the cores and LHCs of PSs [56,63]. Such results are oppositely confirmed in *P*. *australis*, which was able to maintain this index stable only at the moderate saline condition of 15 g L^−1^ NaCl, suggesting a tolerance capacity of this species to relatively low salt concentrations (>250 mM NaCl) and a higher sensitivity to saline and hypersaline conditions than the other halophytes tested. In this context, *J*. *maritimus* occupied an intermediate position, with a progressively continuous and strong reduction of Chl *a*/*b* under the highest NaCl concentrations of 50 and 100 g L^−1^.

The pattern of carotenoid content in leaves and shoots of the four halophytes was very similar to that of chlorophylls, with a higher basic amount and a more gradual decrease with the increasing NaCl concentrations in the two dicot species *S*. *fruticosa* and *H*. *strobilaceum* than the monocots *J*. *maritimus* and *P*. *australis*. The amount and stability of carotenoids in photosynthetic organs of plants is particularly important under stress because these pigments are not only structural components of the photosynthetic apparatus, but they also take part in the light-harvesting processes and assembly of the PSII complex, modulating the integrity of membranes, and protecting thylakoids from environmental stresses [56,67]. In this regard, in *H*. *strobilaceum* treated with 30 g L^−1^ NaCl, it is worth mentioning the 40% increase in leaf carotenoid content, again reaching control levels after a decrease during the first three weeks of treatment. The succulence and the C4 photosynthesis pathway of *S*. *fruticosa* [43,68] and the high carotenoid content and stable distribution of Chl *a* and Chl *b* in the photosynthetic apparatus of *H*. *strobilaceum* [41] are probably involved in tolerance mechanisms by which these halophytic species counteract salt stress under saline and hypersaline conditions.

### 3.3. High Salinity Levels Affect the Distribution of Nutrients among Plant Organs

Treatments with NaCl affected the uptake and accumulation of a range of elements in different organs of the four halophytic species investigated. Among macronutrients, N and P are strictly related to the plant’s assimilation, growth, photosynthesis, reproduction capacity, and energy-producing processes, while K is the most abundant cation involved in many enzymatic reactions and ionic and pH homeostasis for the maintenance of adequate membrane potential [69,70]. The concentrations of such elements in plants, including halophytic and xerophytic species, is adversely influenced by water and soil salinization, which limits plant nutrient access and growth [71,72,73]. In our work, the N content and distribution between stem/rhizome and roots in response to NaCl was similar within the two groups of dicotyledonous and monocotyledonous species, while it differed between these two groups. In *S*. *fruticosa* and *H*. *strobilaceum*, N was unaltered in the stem and decreased in the root between 50 and 100 g L^−1^ NaCl; in *J*. *maritimus* and *P*. *australis*, the N allocation in the rhizome was inhibited only at 100 g L^−1^ NaCl, while in the root, the N uptake progressively reduced with the increasing NaCl concentration. This behavior reveals the different tolerance strategies adopted by the two groups of halophytes in relation to their ecology and morphology. Indeed, in the two dicotyledonous plants, the stem showed no changes in N utilization or protein metabolism because the main function of this organ is to transport absorbed water and nutrients to leaves, while in the two monocotyledonous plants the rhizome, used for the storage of reserves and vegetative reproduction, modulates changes and eventual impairments between shoot and root by remobilizing elements or stimulating their uptake [44,73]. Nitrogen can alleviate salt-induced damage in plants by mediating the utilization of P, K, and other elements and by serving as a fundamental component of proteins involved in a series of metabolic processes to coordinate plant growth and development [71,72]. As expected, the most relevant N reductions found in the organs of all the treated halophytes investigated corresponded to the major inhibitions of biomass described at the highest NaCl doses in *S*. *fruticosa* and *H*. *strobilaceum* and the two monocotyledonous plants, starting from 30 g L^−1^ NaCl. In the two rhizomatous plants, *P*. *australis* and *J*. *maritimus*, in particular, the lower N content occurred with the reduction of the above:below-ground ratio in favor of below-ground (roots and rhizomes) biomass, which was registered at 30, 50, and 100 g L^−1^ NaCl. This latter is a typical growth adjustment observed in such species under low-nutrient or stress conditions [44,47,71].

In the two euhalophytes *S*. *fruticosa* and *H*. *strobilaceum*, the P concentration was enhanced in all plant organs in the presence of the highest NaCl dose (100 g L^−1^), while it was subjected to modest changes in leaves and increased or remained unchanged in the stem and root. These modifications indicated an active uptake and/or translocation of P from roots to leaves, where it supports the energy requirements of photosynthesis. Phosphorous variations in the organs of the two monocot halophytes again suggested a pivotal role of the rhizome as a storage and remobilization organ for reserves in *J*. *maritimus*, while this function was probably inhibited by NaCl in *P*. *australis* that was not able to compensate for the P request by root uptake because it was invested by severe stunting and tissue damages. In any case, rather than absolute N and P values, the N:P ratio in the plant seems to be important because it can reflect the dynamic balance between plant nutrition requirements and the substrate availability of the two elements, influencing the growth and reproduction of individual plants [72,74]. In short-term experiments at the vegetation level, an N:P ratio < 10 and >20 often corresponded to N- and P-limited biomass production and metabolic/nutritional imbalance, although depending on species, growth rate, plant age, and plant parts [72,75]. This is consistent with our findings, since, for example, the N:P ratios of the aerial parts of the two dicot halophytes were always in the range 10–20, while it was <10 in the shoot of *J*. *maritimus* (7.3) under 100 g L^−1^ NaCl, and in the shoot of *P*. *australis* (7.1–7.9) treated with NaCl (Appendix A).

With Ca and Mg, K is the main abundant cation in the plant cell. The hydrated form of the monovalent K^+^ is chemically and structurally very similar to that of Na^+^, and some biophysical roles of K, particularly turgor generation, can be fulfilled by Na [70,76,77]. For this reason, many studies have been conducted on the selective uptake, transport, and accumulation of K over Na, and the tissue distribution of K and Na in combination with the K/Na ratio are generally considered key determinants of plant tolerance to salinity [50,76,78]. Here, the leaf K content of *S*. *fruticosa* strongly increased with the enhancement of NaCl treatment while tending to decrease in the stem and root, as evidenced by an active translocation of this element from roots to shoots. In *H*. *strobilaceum*, the K concentration was maintained at a stable level in the leaves and stem while decreased in the roots up to 30 g L^−1^ NaCl, when the K influx in leaf and stem tissues reached maximum levels and then dropped to the minimum at 100 g L^−1^ NaCl. This different behavior of the two dicot halophytes analysed can be explained by the findings that many halophytes, especially those leaf succulents such as *S*. *fruticosa*, can accumulate high concentrations of K in response to saline conditions, but K cannot substitute for Na in all halophytes as reported for other members of the *Amaranthaceae* [79,80]. Moreover, the osmolyte cell function of K can also be substituted by other inorganic ions, such as Ca^2+^ or Mg^2+^ [38,50,78]. In this respect, it is particularly worth mentioning the high concentrations of both Ca and Mg measured in the shoot and root of *H*. *strobilaceum* and their increase with the enhancement of external NaCl, even at the highest treatment of 100 g L^−1^ NaCl. In *J*. *maritimus*, the shoot and rhizome content of K started to enhance from 30 g L^−1^ NaCl, probably to contrast the high and progressive Na^+^ enter in the root and rhizome. On the other hand, in *P*. *australis*, the K contents of shoot and root progressively decreased with the increasing of NaCl treatments while accumulating at high levels (>5-fold higher) in the rhizome. The accumulation of K in the rhizome of *P*. *australis* at the highest NaCl dose of 100 g L^−1^, together with the high concentrations of Ca and Mg, could be a survival strategy of these perennial plants with deciduous leaves and an extensive rhizome system to overcome salt stress by preparing for the next growing season [50,72,73]. However, increased levels of NaCl in the environment create osmotic and water stress in plants but, at the same time, provide a reduction of cell osmotic potential, hence preventing water loss and enhancing the absorption and transport of ions [13,33]. For these reasons, rather than the absolute concentrations of K, Na, or Ca in plant tissues, changes in the K/Na and Ca/Na ratios become more important, indicating the plant’s ability to maintain the balance of nutrients in the presence of high external Na levels with a low degree of damage [13,50,78]. For this reason, values of K/Na and Ca/Na ratios close to or over 1.0 in the different organs of treated plants showed a general, high ability of all the halophytes to control and adjust the K and Ca concentrations in leaves and shoots, balancing the Na uptake and translocation. In this respect, K seemed to give a higher contribution than Ca to the potential competitive interaction with the accumulation of Na in all the four halophytes tested, probably also thanks to its higher affinity to the monovalent cation [76,77]. In *S*. *fruticosa* and *H*. *strobilaceum* the contribution of K to osmotic adjustment in response to hypersaline NaCl levels was enhanced by the concentrations of both Ca and Mg.

### 3.4. Sodium Accumulation and Distribution Pattern

In plant under salt stress, the osmotic adjustment initiates relatively quickly, and cellular ion homeostasis is predominantly maintained by the accumulation of organic compounds used as compatible solutes or osmolytes [7,13]. Unfortunately, the biosynthesis of these osmolytes is a high cost for the plant, whereas the same cellular osmolarity can be achieved by ion uptake and transport with much lower energy consumption. Indeed, in all salt-tolerant plants, including many halophytes, the osmotic adjustment is also reached through the accumulation of inorganic ions, such as Na^+^ and Cl^−^, which are primarily stored in the vacuole [13,33]. This seems particularly evident in dicotyledonous succulent euhalophytes, which possess swollen vacuoles able to accumulate high amounts of Na^+^ and Cl^−^ in addition to other inorganic and organic solutes; thus, in these species, Na content is naturally high, independently of the NaCl external concentration. In this respect, in *Chenopodiaceae* species, Na^+^ and Cl^−^ mainly represent 67% of the solute concentration (molar in shoot water), while in the *Poaceae*, the same ions averaged only 32% of the solutes [71,81]. This is consistent with results found in the two dicotyledonous halophytes *S*. *fruticosa* and *H*. *strobilaceum*, in which the Na concentration and content in the leaves were basically high in control plants, showing the highest Na levels measured in these organs compared to the other species analysed and maintaining such high Na levels even in the presence of NaCl treatments. Moreover, *S*. *fruticosa* is a halophyte with succulent leaves composed of enlarged cells in which the vacuoles occupy most of the volume and where Na^+^ and Cl^−^ reach high concentrations to prevent dehydration of the cytoplasm [43,68,79]. Other authors have found high Na^+^ levels in *S*. *fruticosa* shoot, which remained unchanged or increased with enhanced salinity [35,82]. The same mechanism seems to be adopted by *Halocnemum* spp., most of which are stem succulents with characteristic shoots, consisting of free bracts with connate opposite small leaves and articulated stems [18,43]. Hence, also in the stem part of the shoot in *H*. *strobilaceum*, the large cells are associated with a decrease in surface area per tissue volume but with a high water content per surface unit [43,81]. Such a morphological trait enables the plant to better regulate the internal ion amounts, maintaining high and constant the translocation and accumulation of Na in the shoot, despite the high external concentrations of 30, 50, and 100 g L^−1^ NaCl tested. Few works reported the Na content of *H*. *strobilaceum* [38], while most studies correlate the spatial distribution of this species with soil or ecosystem salinity, e.g., [18,20,21,39,42], and, to our knowledge, none of these studies describes the cation and sodium contents in different organs as in this work. Both in *S*. *fruticosa* and *H*. *strobilaceum*, the Na concentrations of the stem and root progressively increase with increasing NaCl treatment, reflecting the Na uptake and translocation capacity of these organs.

In the two monocot and rhizomatous halophytes *J*. *maritimus* and *P*. *australis*, the Na concentration and content in different organs was basically and relatively lower than in *S*. *fruticosa* and *H*. *strobilaceum*. However, when exposed to high and hypersaline NaCl concentrations, both *J*. *maritimus* and *P*. *australis* responded in a NaCl dose-response manner, with the only exception of *J*. *maritimus* leaves, in which the Na accumulation was higher at 50 than at 100 g L^−1^ NaCl. The positive and progressively increased Na amounts in leaves, stems, and roots with enhancing NaCl treatments (100–500 mM) were previously found both in *J*. *maritimus* and *P*. *australis* [45,52]. This seems to be the common behavior of most halotolerant species that accumulate Na in their tissues with increasing external NaCl concentrations until a threshold is reached at which these species activate exclusion mechanisms, blocking the transport of toxic ions from the roots to the aerial parts [8,13]. Indeed, although the Na content in both root and rhizome of *J*. *maritimus* and *P*. *australis* increased with the increasing of NaCl dose, the Na content in their shoots remained low, showing a limited translocation capacity from the root.

In our experiment, the total Na amount in the aerial part of *S*. *fruticosa* and *H*. *strobilaceum* was on average about 11-fold higher than in the monocot *J*. *maritimus* and *P*. *australis*, regardless of the external NaCl concentration. It is true that after four weeks of NaCl treatments, the total Na content found in the plants of *J*. *maritimus* subjected to 30, 50, and 100 g L^−1^ NaCl, and in the plants of *P*. *australis* under 100 g L^−1^ reached the same Na amounts removed by the two dicotyledonous plants *S*. *fruticosa* and *H*. *strobilaceum*; however, in such conditions, *J*. *maritimus* and *P*. *australis* show severe physiological impairments and an unbalanced distribution of nutrients with visible damage. Moreover, while for *S*. *fruticosa* and *H*. *strobilaceum* almost 90% of the Na removed was, on average, contained in the aerial part, in *J*. *maritimus* 44% of Na was absorbed by the roots, and in *P*. *australis* 73% of Na was stored in the rhizome. These results obtained in the two monocot halophytes suggest a protective mechanism of the photosynthesizing aboveground tissues with respect to the belowground organs (rhizome and root), which absorbed the excess Na, potentially toxic, sequestering it in their persistent and less active tissues [44,71]. These different Na accumulation/distribution capacities and salt tolerance strategies of the two dicots versus the two monocotyledonous halophytes analysed are confirmed and evidenced by variations in Na bioaccumulation and translocation factors (BAF and TF) and by the opposite (positive in dicots and negative in monocots) correlation coefficients between Na leaf-shoot content and the most physiological/biochemical attributes. In particular, in both *S*. *fruticosa* and *H*. *strobilaceum*, the Na BAF in the shoot was more than one under all the NaCl treatments except at 100 g L^−1^ NaCl (0.5–0.6), and, in any case, the TF was more than one and comparable to or higher than the BAF of shoot and whole plant. These values of BAF and TF are usually used for identification and classification of plants that are accumulators and hyperaccumulators of metals and salts [8,83,84]. On the other hand, in *J*. *maritimus*, the Na BAF in the shoot was higher than one only at the lower NaCl dose (15 g L^−1^, 257 mM), while in *P*. *australis* it was always less than one, and in both species, the TF was always under one, revealing a scarce translocation capacity of Na from the belowground to the aboveground organs. Again, these results showed a higher tolerance to salinity and hypersalinity of the two dicotyledonous plants compared to the two monocots, as well as their higher capability of Na absorption and translocation to the shoot with possible application in phytodesalinization processes.

## 4. Materials and Methods

### 4.1. Plant Material Characteristics

The halophytic species *Suaeda fruticosa*, *Halocnemum strobilaceum*, and the two rhizomatous species *Juncus maritimus* and *Phragmites australis* were chosen due to their adaptability to salinity and diffusion in the Mediterranean area (Table 1). These species developed in natural habitats, such as wetlands or near the coast, generally subjected to floods of fresh water where the level of salinity is not constant. In addition, the choice was also driven by the secondary economic benefits of these halophytes, such as easy cultivation, potential reuse as valuable material, and commercial availability at nurseries [9].

### 4.2. Growth Conditions: Hydroponic Floating Test

For each species, plants of the same age (about six months) and height were purchased from a commercial nursery located in Cagliari (Sardinia, Italy), which selected them from local salt marshes. All plant species were vegetatively propagated and supplied in phytocells filled with organic soil. The experiment was carried out in a greenhouse under semi-controlled conditions: 16–30 °C daytime temperature range, 50–60% relative humidity (RH), 100 ± 40 µmol m^−2^ s^−1^ average of photosynthetic active radiation (PAR) (10.00 a.m.–5.00 p.m.), and 430 ± 60 mg kg^−1^ CO_2_ concentration. During the transfer of plants from pots to hydroponics, the soil was carefully removed from the roots, which were thoroughly washed with running tap water. The experimental set up consisted of mesocosms (plastic containers with a size of 19 × 26 × 15 cm) filled with 3 L of hydroponic solution and containing two plants. The hydroponic solution was prepared with a commercial complete liquid fertilizer, COMPO Italia S. r. l., diluted in tap water. The derived nutrient solution contained nitrogen (N), potassium (K_2_O), and phosphorus (P_2_O_5_) in the 7:3:6 ratio. The micronutrients copper (Cu), iron (Fe), manganese (Mn), and zinc (Zn) were added both in the water soluble and EDTA chelated forms. For macro- and micronutrients, the use of chemical products containing chlorine (Cl) was avoided to reduce the potential toxic effects of this element and possible additional effect on NaCl treatments. A polystyrene plate was fitted at the lip of each mesocosm to provide physical support for plants and to reduce the evaporation of the solution. Each plant, at the root collar point, was inserted into a special hole in the polystyrene plate, where a plastic support kept the plant erect; the roots were completely free and floating in the hydroponic solution without any type of substrate [85]. The aeration of the system was ensured by aquarium pumps (250 L h^−1^) and by the weekly renewal of nutrient solutions, as previously described [86]. The staff provided 7–12 ppm of dissolved oxygen in all solutions. After acclimation to hydroponic conditions, the four plant species were randomly distributed to the following NaCl concentrations for four weeks (28 days): 15, 30, 50, and 100 g L^−1^, corresponding to 257, 514, 856, and 1712 mM NaCl, respectively. Control plants were grown under the same conditions, but without (0 g L^−1^) NaCl. To avoid osmotic shock, the NaCl addition was carried out gradually, with a daily (2.5–5 g L^−1^ NaCl) application of salt until the desired concentration was reached. Two mesocosms with two plants of the same species were used for each species and salt treatment, for a total of 40 mesocosms (scheme of Figure 8). The hydroponic solution (pH = 5) was replaced every 7 days, keeping the same salt and pH conditions established to prevent significant variations in osmotic potential.

### 4.3. Growth Indicators and Hydration

Throughout the experiment, the plants were evaluated daily for a visual rating of stress symptoms (chlorosis spots, wilting, crumpled leaves). At harvest (28 days after NaCl treatments), leaves, stem, and root (in the case of *S*. *fruticosa* and *H*. *strobilaceum*) and shoot, rhizome, and root (in the case of *J*. *maritimus* and *P*. *australis*) of each plant were separated, washed with tap water, and then rinsed with deionized water and blotted between two layers of filtered paper. The fresh weight (FW) of each plant material was recorded immediately before washing, and after drying in an oven at 60 °C to a stable weight, the dry weight (DW) was determined. The water content of each plant organ analysed was measured using the following formula as a percentage of FW and denominated relative water content (RWC) [78]:RWC (%) = (FW − DW)/FW × 100

This parameter was used to estimate the relative water content of control plants and organs in comparison with those under NaCl treatment conditions.

The root to shoot ratio, the shoot (SMR), and the root (RMR) to total biomass ratios were calculated [86].

### 4.4. Chlorophyll Fluorescence Parameters

Light and dark-adapted fluorescence parameters were measured before the beginning of NaCl treatments and every 7 days from the start of the treatments until plant harvesting using a portable miniaturized pulse-amplitude-modulated fluorimeter (Mini-PAM; Heinz WalzGmbH, Effeltrich, Germany). Measurements (10–12 a.m.) were made on each of the four plants for each species and treatment (*n* = 4), according to the protocol previously reported [57]. For *S*. *fruticosa*, *P*. *australis* and *J*. *maritimus*, fully-expanded and light-exposed leaves were used, while for *H*. *strobilaceum* measurements were conducted on the scale-like leaves clasping the stem of the upper branchlets. The photosynthetic photon flux density (PPFD) of the saturation pulses to determine the maximal fluorescence emission in the absence (Fm) and the presence (Fm′) of actinic light was about 8000 µmol m^−2^ s^−1^. Fluorescence parameters were determined at both growing PPFD (~100 µmol m^−2^ s^−1^) and after acclimation to dark conditions with leaf clips for at least 30 min. The potential efficiency of photosystem II (PSII) photochemistry (Fv/Fm) was calculated on dark-adapted leaves as follows:Fv/Fm = (Fm − Fo)/Fm
where Fv is the variable chlorophyll fluorescence, and Fo and Fm are the minimal and the maximum fluorescence yields emitted by the leaves in the dark-adapted state, respectively. The actual efficiency of PSII photochemistry in the light (ΦPSII) was calculated at growing light intensity when steady state was achieved as follows:ΦPSII = (Fm′ − F′)/Fm′
where Fm′ is the maximum fluorescence yield with all PSII reaction centers in the reduced state, obtained by superimposing a saturating light flash during exposition to actinic light, and F′ is the fluorescence at the actual state of PSII reaction centers during actinic illumination. The fast-relaxing component of non-photochemical fluorescence quenching (NPQ) was estimated according to the Stern-Volmer equation [87]:NPQ = Fm/Fm′ − 1

### 4.5. Photosynthetic Pigments

Photosynthetic pigments (chlorophyll *a*, Chl *a*; chlorophyll *b*, Chl *b*; carotenoids) in the leaves of the four halophytes were detected according to the modified acetone extraction method described by [88] on samples taken at the same time for the measurement of the fluorescence parameters. Leaf or shoot samples (about 0.1 g FW) were collected from each plant species, packed in an aluminum sheet, frozen in liquid nitrogen, and then stored at −80 °C. Subsequently, the samples were homogenized in 80% (*w*/*v*) cold acetone and centrifuged at 12,000 rpm for 10 min at 4 °C. The supernatant obtained, eventually filtered (0.2 μm, Sartorius Stedim Biotech, Göttingen, Germany), was used for the analyses. The absorbance of these extracts was measured at 663.2, 646.8, and 470.0 nm with an UV-vis spectrophotometer (UV-1800 Spectrophotometer, Shimadzu, Kyoto, Japan).

### 4.6. Chemical Tissue Analysis: Major Element Distribution; Na Transport and Accumulation in Different Plant Organs

The nutrients nitrogen (N), phosphorous (P), potassium (K), calcium (Ca), magnesium (Mg), and sodium (Na) were determined in the leaves, stem or shoot, rhizome, and root of each plant species for each experimental condition. The N content was determined in approximately 4 mg of dried and pulverized vegetal samples using an elemental analyzer (Model NA 1500, Carlo Erba, Milan, Italy). The oven-dried plant material (0.5 g) was digested in a solution of nitric acid (HNO_3_ 65%) and hydrogen peroxide (H_2_O_2_ 30%) (2:1, *v*/*v*) in a closed-vessel microwave-assisted digestion system (Milestone Ethos 900, Bergamo, Italy) using US-EPA Method 3052 [89]. The content of K, Mg, and Ca was analyzed by inductively coupled plasma optical emission spectroscopy (ICP-OES 5900, Agilent, Santa Clara, California, USA), and the Na concentration was determined using atomic absorption spectrometry (AA240 FS, Agilent Technologies, Santa Clara, California, USA). The content of elemental P was determined following the molybdate blue ascorbic acid method [90]. When the weight of plant material was not sufficient for chemical determinations, samples from the same species, organ, and treatment were bulked, reaching the weight for at least two extractions (*n* = 2). The capacity of plants in taking up Na and then translocating it from roots to the other organs was evaluated via examination of both bioaccumulation factor (BAF) and translocation factor (TF), and total removal (TR) or content [83,84] in the different plant species, according to the following equations:BAF = Na_plant organ_/Na_medium_
TF = Na_shoot_/Na_belowground_
where Na_plant organ_, Na_shoot_, Na_belowground_, and Na_medium_ are the content (mg per plant) of sodium in the plant organ considered, in the shoot, in the belowground parts (root or rhizome + root) and in the nutrient solution, respectively. The TR, or content of Na in plant organs and the whole plant, was calculated by multiplying the Na concentration (mg g^−1^) in each plant organ for the correspondent DW and summing the products. The relative contents of K or Ca over Na were evaluated by the K or Ca concentration to Na concentration ratios (K/Na, Ca/Na) in different plant organs.

### 4.7. Statistical Analysis

The experiments were arranged in a complete randomized design with different NaCl treatments and plant species by at least three replicates (*n* = 3), or as otherwise indicated in tables and figures. The NaCl treatment factor (NaCl) had five levels (0 g L^−1^ NaCl addition, control; 15, 30, 50, and 100 g L^−1^ NaCl). The species factor (S) was represented by the four selected halophytes: *S*. *fruticosa*, *H*. *strobilaceum*, *J*. *maritimus* and *P*. *australis*. For the evaluation of these two factors and their interaction, a two-way analysis of variance (two-way ANOVA) was applied to the results. For non-destructive analyses (chlorophyll fluorescence and photosynthetic pigments) measured every week (7, 14, 21, and 28 days after treatment), a one-way ANOVA was performed to test significant differences among NaCl treatments within each species and sampling time. All datasets were checked for the fulfilment of ANOVA assumptions (robust Levene’s test of homogeneity of variances). In any case, the separation of means was performed by using the Fisher’s least significance difference (LSD) test with a significance level (*p*) ≤ 0.05. For each species, Pearson correlation and/or linear regression coefficients were calculated to reveal relationships among the different variables in response to increasing NaCl concentrations. All statistical analyses were conducted using the STATISTICA 8.0 software package (StatSoft Inc., Tulsa, OK, USA).

## 5. Conclusions

Based on the results of this halophyte screening under high and hyper-saline conditions, we can conclude that the four species investigated adopted different behaviors and tolerance strategies to counteract the NaCl stress in dependence of their species-specific characteristics, life form, salt concentration, and time of exposure. To our knowledge, this is the first experiment conducted in controlled hydroponic conditions in which mature plants of four different halophytes are exposed to high and hypersaline NaCl concentrations, simulating the saline levels of marine or produced water from industrial setups and of arid and semiarid regions of the world characterized by seawater intrusion or excessive salt accumulation by irrigation.

The two dicotyledonous species, *S*. *fruticosa* and *H*. *strobilaceum*, were more tolerant of hypersaline concentrations (30, 50, and 100 g L^−1^ NaCl) than the two monocot rhizomatous species, *J*. *maritimus* and *P*. *australis*, and between these, *P*. *australis* was more sensitive to salt stress than *J*. *maritimus*, showing severe injury symptoms and a reduction in vitality since the first two weeks of treatment.

Morphophysiological traits revealed a more efficient capacity of *S*. *fruticosa* and *H*. *strobilaceum* to counteract the osmotic impairments and nutrient unbalances induced by the exposure to hypersaline concentrations. However, in all four halophytic species, the differential absorption (and/or the storage in the rhizome) of the major cations K, Ca, and Mg seemed to mitigate or partially counterbalance the effects of the high Na amounts.

The two euhalophytes, *S*. *fruticosa* and *H*. *strobilaceum*, absorbed and accumulated high Na amounts in the shoot at all the NaCl doses tested, reaching values of BAF and TF only found in hyperaccumulator plants of metals. In the two rhizomatous monocotyledons, Na progressively increased with the increasing concentration of external NaCl, but the growth in *J*. *maritimus* was dramatically inhibited at 50 and 100 g L^−1^ NaCl and the vitality of *P*. *australis* was compromised from 30 g L^−1^ NaCl after two weeks of treatment.

These results offer useful findings for the selection and use of halophytic species in restoration and revegetation of hypersaline environments and/or remediation of high-saline industrial PW and suggest practical applications for phytomanagement of hypersalinity. For example, *S*. *fruticosa* and *H*. *strobilaceum* can be used in environments or wastewaters with extremely high NaCl concentrations or for longer time periods than *J*. *maritimus* and *P*. *australis*. In any case, additional fertilization with K, Ca, and/or Mg may be applied to ameliorate plants’ performance. Halophytes tolerant to hypersaline conditions do not compete for the resources required by crops in conventional agriculture; thus, their cultivation can be sustainable not only for desalinization practices but also for the recycling of plant material in feedstock or energy production.

Further studies and screening tests are necessary to identify the tolerance mechanisms of halophytes to hypersaline conditions and their resilience and recovery capacity, depending on the specific environmental application. In this respect, the tests proposed here are useful for fast screening and identification of salt hyperaccumulators among halophytic species. Ad hoc experiments to evaluate halophytes response to and degradation capacity for eventual organic contaminants present in hypersaline conditions are in progress, comparing the two most promising species, *S*. *fruticosa* and *H*. *strobilaceum*, for tolerance to extreme saline conditions. Moreover, the contribution of associated microrganisms (endophytic bacteria) to the growth and resistance of halophytes to hypersaline environments will represent an interesting area of investigation.

## Figures and Tables

**Figure 1 plants-12-01737-f001:**
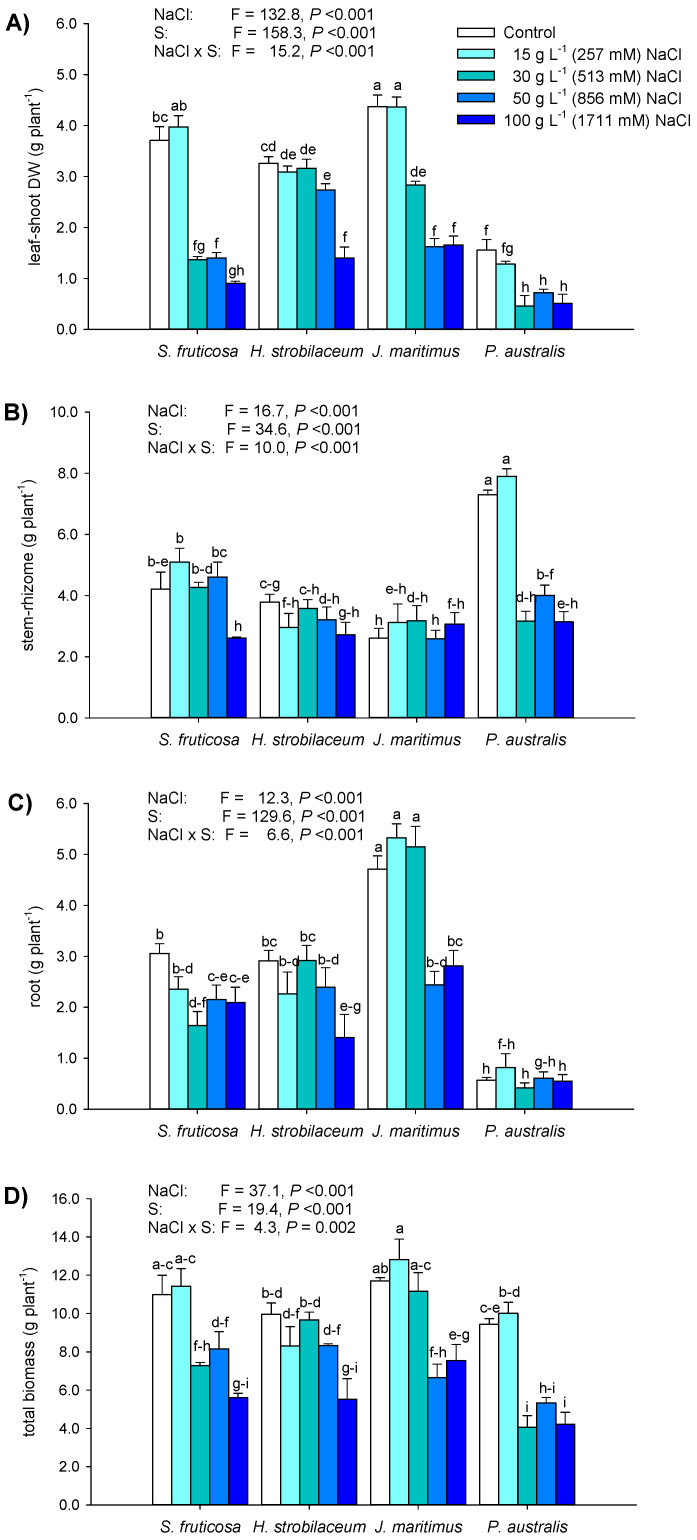
Biomass partitioning (DW, g plant^−1^) in different organs (leaf or shoot, (**A**); stem or rhizome, (**B**); root, (**C**); whole plant, (**D**)) of the four halophyte plant species treated with 0 (control), 15, 30, 50, and 100 g L^−1^ NaCl (corresponding to 0, 257, 514, 856, and 1712 mM NaCl, respectively) for 28 days. Values are the means ± standard error (SE) of four plants (*n* = 4). Results of the two-way ANOVA (*p* ≤ 0.05) for the effect of NaCl treatments and species and of their interaction are shown (F and *p* values). When the interaction between factors was significant, the Fisher LSD-test (*p* ≤ 0.05) was applied: significantly different data are followed by different letters in the histogram columns of the same graph. DW, dry weight.

**Figure 2 plants-12-01737-f002:**
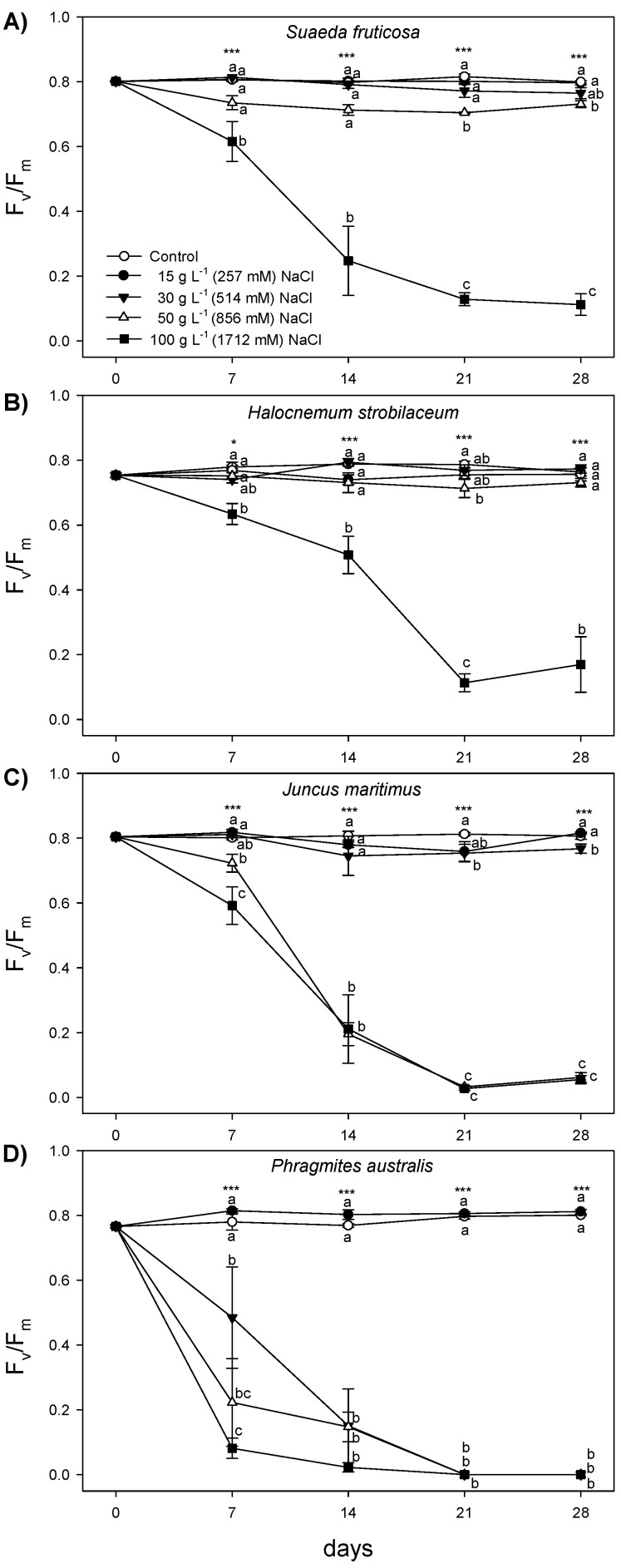
Maximum PSII photochemical efficiency (Fv/Fm) (dark-adapted leaves) in *S*. *fruticosa* (**A**), *H*. *strobilaceum* (**B**), *J*. *maritimus* (**C**), and *P*. *autralis* (**D**) exposed for 28 days to increasing NaCl concentrations (0—control, 15, 30, 50, and 100 g L^−1^, corresponding to 0, 257, 514, 856, and 1712 mM, respectively). Data reported in the graphs are the mean values ± SE (*n* = 4). For each time point, the results of a one way-ANOVA (NaCl-treated plants against the controls) are indicated (*, *p* ≤ 0.05; ***, *p* ≤ 0.001). Different letters correspond to significant differences for the Fisher’s LSD post-hoc test (*p* ≤ 0.05).

**Figure 3 plants-12-01737-f003:**
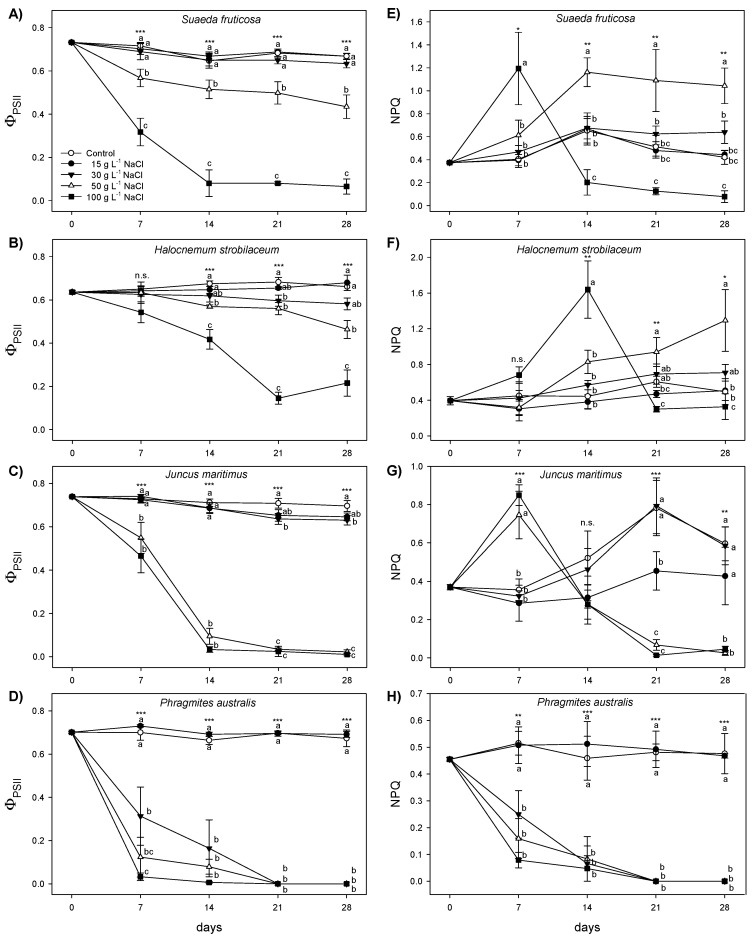
Actual efficiency of PSII photochemistry in the light (ΦPSII) and non-photochemical fluorescence quenching (NPQ) in *S*. *fruticosa* (**A**,**E**), *H*. *strobilaceum* (**B**,**F**), *J*. *maritimus* (**C**,**G**), and *P*. *autralis* (**D**,**H**) exposed for 28 days to increasing NaCl concentrations (0—control, 15, 30, 50, and 100 g L^−1^, corresponding to 0, 257, 514, 856, and 1712 mM, respectively). Data reported in the graphs are the mean values ± SE (*n* = 4). For each time point, the results of a one way-ANOVA (NaCl-treated plants against the controls) are indicated (*, *p* ≤ 0.05; **, *p* ≤ 0.01; ***, *p* ≤ 0.001; n.s., not significant). Different letters correspond to significant differences for the Fisher’s LSD post-hoc test (*p* ≤ 0.05).

**Figure 4 plants-12-01737-f004:**
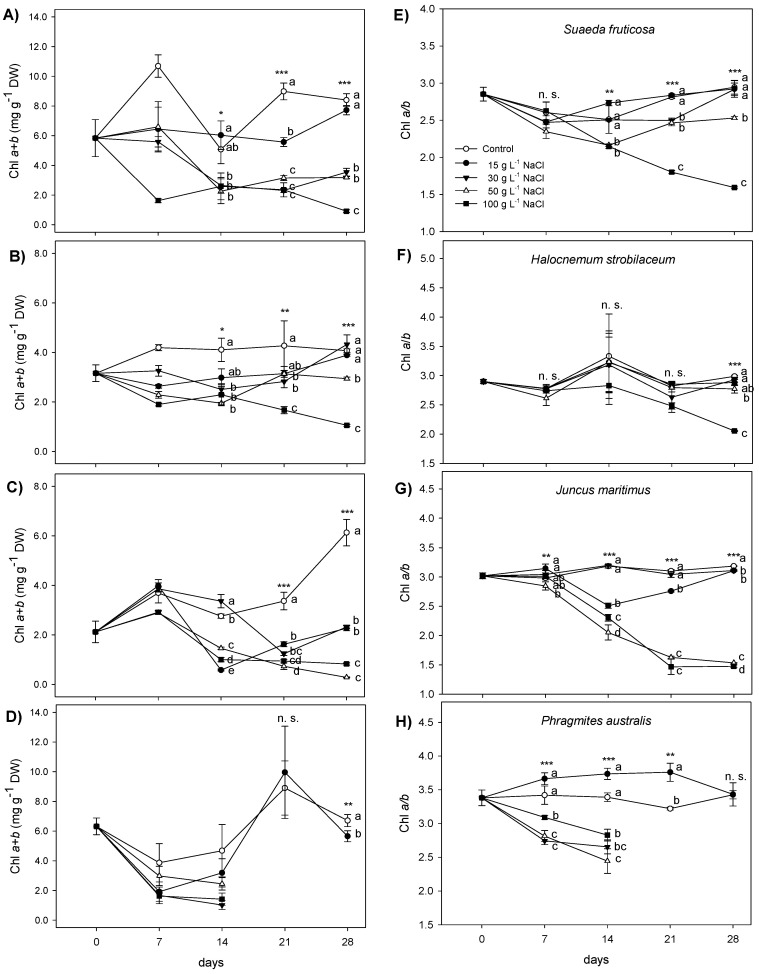
Total chlorophyll (Chl *a* + *b*, mg g^−1^ DW) and the chlorophyll *a* to *b* ratio (Chl *a*/*b*) in leaves of *S. fruticosa* (**A**,**E**), *H*. *strobilaceum* (**B**,**F**), *J*. *maritimus* (**C**,**G**), and *P*. *autralis* (**D**,**H**) exposed for 28 days to increasing NaCl concentrations (0—control, 15, 30, 50, and 100 g L^−1^, corresponding to 0, 257, 514, 856, and 1712 mM, respectively). Data reported in the graphs are the mean values ± SE; each mean refers to three or four extractions or replications derived from leaf material of one plant or bulked material of two plants (*n* = 3–4) for each species and NaCl treatment. For each time point, the results of a one way-ANOVA (NaCl-treated plants against the controls) are indicated (*, *p* ≤ 0.05; **, *p* ≤ 0.01; ***, *p* ≤ 0.001; n.s., not significant). Different letters correspond to significant differences for the Fisher’s LSD post-hoc test (*p* ≤ 0.05).

**Figure 5 plants-12-01737-f005:**
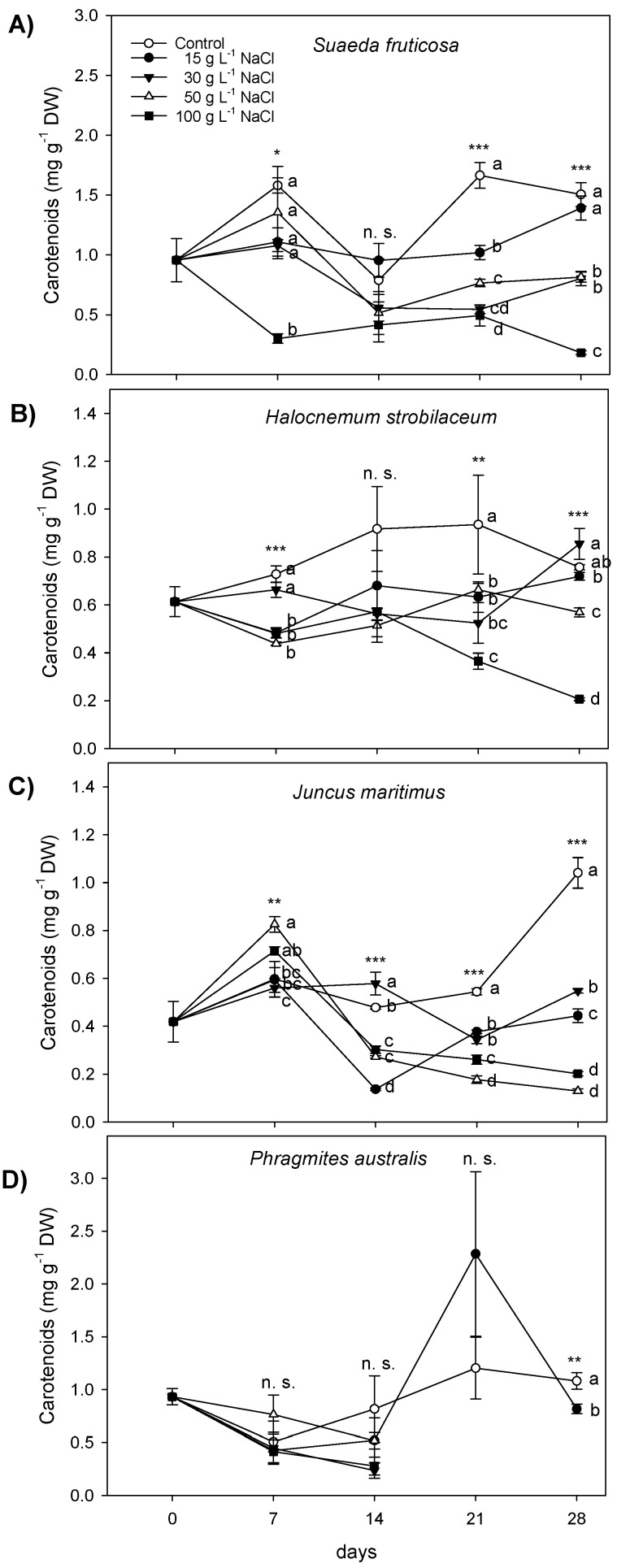
Carotenoids (mg g^−1^ DW) in the leaves of *S*. *fruticosa* (**A**), *H*. *strobilaceum* (**B**), *J*. *maritimus* (**C**), and *P*. *autralis* (**D**) exposed for 28 days to increasing NaCl concentrations (0—control, 15, 30, 50, and 100 g L^−1^, corresponding to 0, 257, 514, 856, and 1712 mM, respectively). Data (mean values ± SE) reported in the graphs refers to three or four extractions or replications derived from leaf material of one plant or bulked material of two plants (*n* = 3–4) for each species and NaCl treatment. For each time point, the results of a one way-ANOVA (NaCl-treated plants against the controls) are indicated (*, *p* ≤ 0.05; **, *p* ≤ 0.01; ***, *p* ≤ 0.001; n.s., not significant). Different letters correspond to significant differences for the Fisher’s LSD post-hoc test (*p* ≤ 0.05).

**Figure 6 plants-12-01737-f006:**
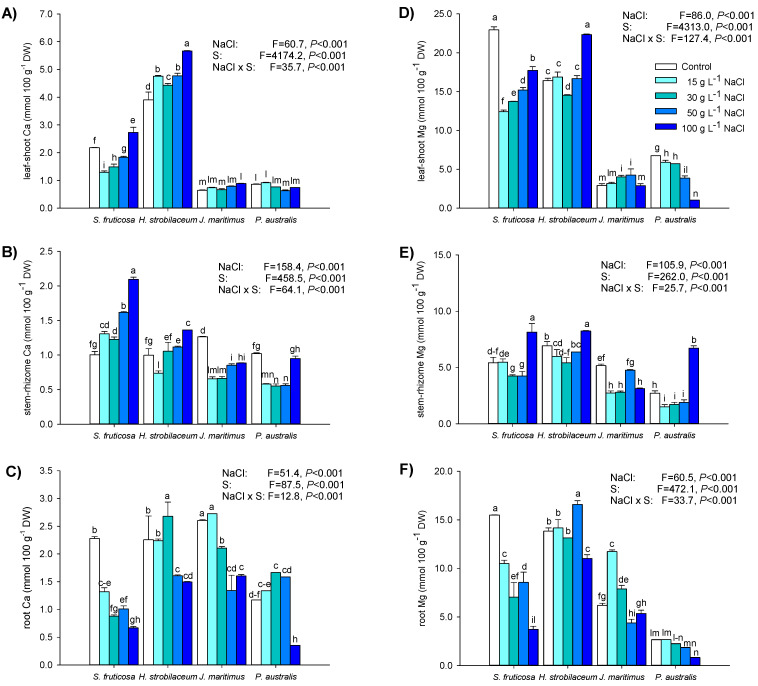
Calcium (Ca) and magnesium (Mg) concentrations (mg g^−1^ DW) in different organs (leaf or shoot—(**A**,**D**); stem or rhizome—(**B**,**E**); root—(**C**,**F**)) of *S*. *fruticosa*, *H*. *strobilaceum*, *J*. *maritimus* and *P*. *australis* treated with 0 (control), 15, 30, 50, and 100 g L^−1^ NaCl (corresponding to 0, 257, 514, 856, and 1712 mM NaCl, respectively) for 28 days. Values are the means ± standard deviation (SD) of two or three extractions (*n* = 2–3) derived from the bulked vegetal material of four plants for each species and NaCl treatment. Results of the two-way ANOVA (*p* ≤ 0.05) for the effects of NaCl treatments (NaCl) and species (S) and of their interaction (NaCl × S) are shown (F and *p* values). When the interaction between factors was significant, the Fisher LSD-test (*p* ≤ 0.05) was applied. Significantly different data are followed by different letters on the histogram columns of the same graph.

**Figure 7 plants-12-01737-f007:**
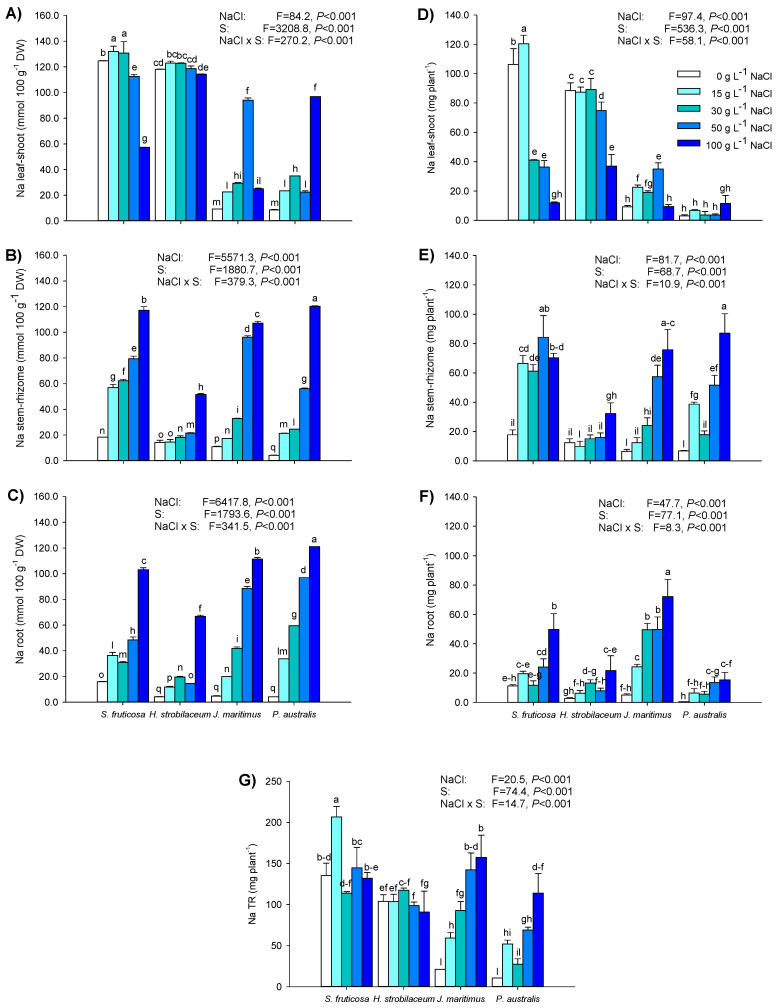
Sodium (Na) concentration (mg g^−1^ DW, (**A**–**C**)) and content or removal (mg plant^−1^, (**D**–**G**)) in different organs (leaf or shoot—(**A**,**D**); stem or rhizome—(**B**,**E**); root—(**C**,**F**); whole plant, (**G**)) of *S*. *fruticosa*, *H*. *strobilaceum*, *J*. *maritimus*, and *P*. *australis* treated with 0 (control), 15, 30, 50, and 100 g L^−1^ NaCl (corresponding to 0, 257, 514, 856, and 1712 mM NaCl, respectively) for 28 days. Values are the means ± SD of two or three extractions (*n* = 2–3) derived from the bulked vegetal material of four plants for each species and NaCl treatment. Results of the two-way ANOVA (*p* ≤ 0.05) for the effects of NaCl treatments (NaCl) and species (S) and of their interaction (NaCl × S) are shown (F and *p* values). When the interaction between factors was significant, the Fisher LSD-test (*p* ≤ 0.05) was applied. Significantly different data are followed by different letters on the histogram columns of the same graph. TR, total removal.

**Figure 8 plants-12-01737-f008:**
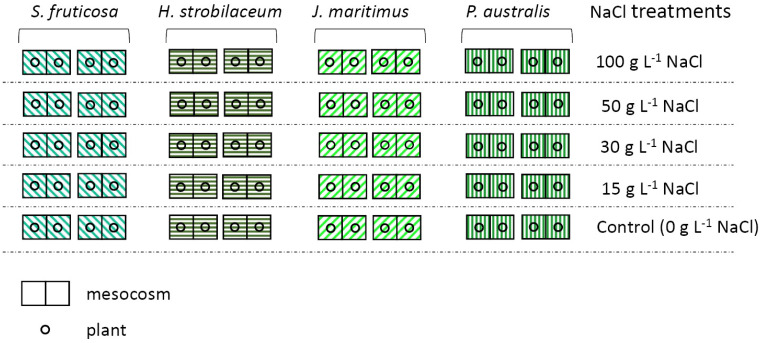
Scheme of the experimental set-up for the hydroponic floating test. For each halophytic species (*Suaeda fruticosa*, *Halocnemum strobilaceum*, *Juncus maritimus*, and *Phragmites australis*), two mesocosms with two plants inside were used for each NaCl treatment. The single plant was the biological replicate, with four replicates per NaCl treatment (*n* = 4), two mesocosms per treatment, and ten mesocosms per species.

**Table 1 plants-12-01737-t001:** Taxonomic and morpho-physiological characteristics of the plant species employed in the research.

Species	Family	Habitat	Life Form and Cycle	Salt Tolerance and Eco-Morphotype	References
*Suaeda fruticosa* (L.) Forssk., syn. *Suaeda vera* J.F. Gmel.	*Amaranthaceae*	Salty marshes of the Mediterranean, Europe, and Africa.	Chamaephyte, perennial	Eu-halophyte, leaf succulent	[21,29,30]
*Halocnemum strobilaceum* (Pallas) M. Bieb., syn. *Salicornia strobilacea* Pall.	*Amaranthaceae*	Salty marshes, and saline soils of the Mediterranean and Asia	Chamaephyte, perennial	Eu-halophyte, stem succulent, xerophytic type	[8,21,30,31]
*Juncus maritimus* Lam.	*Juncaceae*	Juncetea maritimi of Europe, West Asia, and Magreb	Geo (crypto) phyte, perennial	Hydroalophyte	[8,21,30,31]
*Phragmites australis* (Cav.) Trin. ex Steudel, syn. *Phragmites communis* Trin.	*Poaceae*	Phragmitetalia, ubiquitous, in both shallow lowland freshwater, and saline (marshes and swamps) habitats	Therophyte, geophyte, and helophyte—hydrophytes depending on the habitat; perennial	Eury-hygro-halophyte	[21,30,31]

**Table 2 plants-12-01737-t002:** Nitrogen (N), phosphorous (P), and potassium (K) in the leaf or shoot, rhizome, or stem and root of the four halophyte plant species treated with 0 (control), 15, 30, 50, and 100 g L^−1^ NaCl (corresponding to 0, 257, 514, 856, and 1712 mM NaCl, respectively) for four weeks. Values are the means ± standard deviation (SD) of two or three extractions (*n* = 2–3) derived from the bulked vegetal material of four plants for each species and NaCl treatment. Results of the two-way ANOVA (*p* ≤ 0.05) for the effects of NaCl treatments and species and of their interaction are shown. When the interaction between factors was significant, the Fisher LSD-test (*p* ≤ 0.05) was applied: significantly different data are followed by different letters in the same column.

	*Suaeda fruticosa*								
	N (%)			P (mmol 100 g^−1^ DW)			K (mmol 100 g^−1^ DW)		
NaCl	leaf	stem	root	leaf	stem	root	leaf	stem	root
Control (0 g L^−1^)	4.94 ± 0.20 a	1.93 ± 0.11 a	2.03 ± 0.05 bc	10.39 ± 0.08 b	5.22 ± 0.07 f	7.70 ± 0.07 g	2.26 ± 0.08 m	10.92 ± 0.04 a	6.90 ± 0.26 b
15 g L^−1^	4.11 ± 0.17 b	1.63 ± 0.02 a–d	2.07 ± 0.18 bc	6.39 ± 0.14 e	4.81 ± 0.09 g	11.77 ± 0.21 b	18.64 ± 0.24 b	8.77 ± 0.38 c	9.18 ± 0.32 a
30 g L^−1^	3.36 ± 0.14 de	1.85 ± 0.23 a–c	1.82 ± 0.16 c–e	4.87 ± 0.14 g	4.45 ± 0.08 h	9.44 ± 0.01 d	13.32 ± 0.80 e	6.06 ± 0.32 e	5.96 ± 0.50 c
50 g L^−1^	3.58 ± 0.10 cd	1.88 ± 0.21 a–c	1.69 ± 0.13 d–f	6.03 ± 0.06 f	8.52 ± 0.22 b	18.23 ± 0.11 a	11.85 ± 0.60 g	9.61 ± 0.17 b	3.78 ± 0.15 fg
100 g L^−1^	3.78 ± 0.03 c	1.91 ± 0.03 ab	1.35 ± 0.05 g	11.69 ± 0.18 a	11.21 ± 0.01 a	11.11 ± 0.19 c	17.44 ± 0.55 c	4.51 ± 0.15 g	4.08 ± 0.33 f
				*Halocnemum strobilaceum*					
	N (%)			P (mmol 100 g^−1^ DW)			K (mmol 100 g^−1^ DW)		
NaCl	leaf	stem	root	leaf	stem	root	leaf	stem	root
Control (0 g L^−1^)	2.82 ± 0.06 f	1.46 ± 0.10 d–f	1.81 ± 0.06 c–e	4.02 ± 0.21 h	2.85 ± 0.06 m	4.26 ± 0.04 il	14.57 ± 0.13 d	3.61 ± 0.03 h	5.23 ± 0.14 d
15 g L^−1^	2.30 ± 0.09 i	1.27 ± 0.04 e–h	1.74 ± 0.02 d–f	3.10 ± 0.004 1	2.70 ± 0.02 m	4.36 ± 0.09 i	11.74 ± 0.33 g	2.45 ± 0.06 lm	3.47 ± 0.17 gh
30 g L^−1^	2.14 ± 0.05 il	1.57 ± 0.27 b–e	1.61 ± 0.07 d–f	2.58 ± 0.05 m	2.67 ± 0.08 m	4.02 ± 0.20 l	12.19 ± 0.09 fg	3.53 ± 0.07 h	2.72 ± 0.04 i
50 g L^−1^	2.35 ± 0.05 h	1.54± 0.18 c–f	1.47 ± 0.03 fg	2.40 ± 0.20 m	3.56 ± 0.22 l	3.34 ± 0.21 m	19.74 ± 0.15 a	5.66 ± 0.01 f	4.77 ± 0.10 e
100 g L^−1^	2.32 ± 0.06 hi	1.44 ± 0.001 d–f	1.57 ± 0.03 ef	4.83 ± 0.07 g	6.69 ± 0.29 c	5.82 ± 0.15 h	2.11 ± 0.02 m	1.15 ± 0.02 o	1.21 ± 0.01 mn
				*Juncus maritimus*					
	N (%)			P (mmol 100 g^−1^ DW)			K (mmol 100 g^−1^ DW)		
NaCl	shoot	rhizome	root	shoot	rhizome	root	shoot	rhizome	root
Control (0 g L^−1^)	1.61 ± 0.02 n	1.34 ± 0.04 d–g	0.79 ± 0.01 i	3.52 ± 0.41 i	1.95 ± 0.20 n	0.05 ± 0.003 p	5.70 ± 0.01 l	2.60 ± 0.26 il	1.71 ± 0.04 l
15 g L^−1^	1.51 ± 0.03 n	1.37 ± 0.19 d–g	0.65 ± 0.04 il	2.55 ± 0.10 m	3.85 ± 0.06 il	0.14 ± 0.09 op	5.96 ± 0.03 l	2.83 ± 0.02 i	0.96 ± 0.02 nl
30 g L^−1^	1.22 ± 0.04 o	1.10 ± 0.09 g–i	0.58 ± 0.01 lm	4.87 ± 0.04 g	6.38 ± 0.15 de	n. d.	7.17 ± 0.19 i	2.59 ± 0.01 il	2.65 ± 0.27 i
50 g L^−1^	1.52 ± 0.02 mn	1.02 ± 0.14 h–l	0.48 ± 0.02 m	4.24 ± 0.15 h	6.17 ± 0.09 e	0.32 ± 0.04 op	7.41 ± 0.12 i	6.54 ± 0.22 d	1.93 ± 0.03 l
100 g L^−1^	1.33 ± 0.05 m–o	0.84 ± 0.01 l	0.61 ± 0.02 lm	5.88 ± 0.18 f	6.64 ± 0.17 cd	0.42 ± 0.09 o	7.38 ± 0.16 i	5.34 ± 0.19 f	1.96 ± 0.12 l
				*Phragmites australis*					
	N (%)			P (mmol 100 g^−1^ DW)			K (mmol 100 g^−1^ DW)		
NaCl	shoot	rhizome	root	shoot	rhizome	root	shoot	rhizome	root
Control (0 g L^−1^)	3.19 ± 0.04 e	1.27 ± 0.04 e–h	3.59 ± 0.36 a	9.93 ± 0.12 c	4.07 ± 0.09 i	8.86 ± 0.01 e	12.85 ± 0.30 e	2.09 ± 0.04 n	5.81 ± 0.01 c
15 g L^−1^	2.53 ± 0.03 gh	1.39 ± 0.05 d–f	2.19 ± 0.26 b	10.42 ± 0.14 b	3.96 ± 0.16 i	8.07 ± 0.003 f	12.79 ± 0.21 ef	2.64 ± 0.14 il	3.08 ± 0.01 hi
30 g L^−1^	2.29 ± 0.05 i	1.62 ± 0.04 a–d	1.85 ± 0.18 cd	10.23 ± 0.003 bc	6.18 ± 0.26 e	9.44 ± 0.01 d	10.27 ± 0.01 h	2.56 ± 0.07 i–m	1.83 ± 0.01 l
50 g L^−1^	2.00 ± 0.03 l	1.23 ± 0.08 g–h	1.80 ± 0.06 c–e	8.60 ± 0.11 d	4.00 ± 0.07 i	8.30 ± 0.01 f	5.97 ± 0.14 l	2.24 ± 0.11 mn	1.58 ± 0.01 lm
100 g L^−1^	2.59 ± 0.14 fg	0.89 ± 0.001 il	1.09 ± 0.01 h	10.13 ± 0.003 bc	2.65 ± 0.06 m	1.34 ± 0.003 n	1.80 ± 0.01 m	11.06 ± 0.16 a	1.61 ± 0.01 lm
Statistics	Two-way ANOVA—F (*P*)								
NaCl	60.7 (<0.001)	4.5 (0.008)	50.6 (<0.001)	386.1 (<0.001)	673.4 (<0.001)	459.0 (<0.001)	302.9 (<0.001)	272.4 (<0.001)	150.4 (<0.001)
species	803.8 (<0.001)	40.2 (<0.001)	315.9 (<0.001)	3173.9 (<0.001)	914.1 (<0.001)	7371.7 (<0.001)	798.9 (<0.001)	1696.2 (<0.001)	651.9 (<0.001)
NaCl x species	7.1 (<0.001)	3.6 (0.004)	14.3 (<0.001)	154.7 (<0.001)	323.7 (<0.001)	664.5 (<0.001)	598.2 (<0.001)	616.3 (<0.001)	88.6 (<0.001)

**Table 3 plants-12-01737-t003:** Na/K and Na/Ca ratios in the different plant tissues (leaves or shoot, stem, or rhizome and root) of the four halophyte plant species treated with 0, 15, 30, 50, and 100 g L^−1^ NaCl (corresponding to 0, 257, 514, 856, and 1712 mM NaCl, respectively) for 28 days. Values are the mean ± SD of two or three extractions (*n* = 2–3) derived from the bulked vegetal material of four plants for each species and NaCl treatment. Results of the two-way ANOVA (*p* ≤ 0.05) for the effects of NaCl treatments and species and of their interaction are shown. When the interaction between factors was significant, the Fisher LSD-test (*p* ≤ 0.05) was applied: significantly different data are followed by different letters in the same column.

	*Suaeda fruticosa*					
NaCl	K/Na leaf	K/Na stem	K/Na root	Ca/Na leaf	Ca/Na stem	Ca/Na root
Control (0 g L^−1^)	0.11 ± 0.004 n	3.52 ± 0.003 a	2.56 ± 0.08 c	0.24 ± 0.002 l	0.74 ± 0.03 d	1.93 ± 0.05 cd
15 g L^−1^	0.84 ± 0.01 h	0.91 ± 0.002 ef	1.49 ± 0.04 g	0.13 ± 0.001 mn	0.31 ± 0.02 f	0.49 ± 0.004 ef
30 g L^−1^	0.60 ± 0.004 l	0.58 ± 0.02 g–l	1.14 ± 0.07 h	0.15 ± 0.02 m	0.27 ± 0.01 fg	0.38 ± 0.02 ef
50 g L^−1^	0.62 ± 0.02 l	0.72 ± 0.01 gh	0.46 ± 0.002 lm	0.22 ± 0.001 l	0.28 ± 0.01 f	0.28 ± 0.003 ef
100 g L^−1^	1.80 ± 0.06 d	0.23 ± 0.01 no	0.23 ± 0.02 op	0.64 ± 0.04 c	0.24 ± 0.01 f–h	0.09 ± 0.002 ef
	*Halocnemum strobilaceum*					
NaCl	K/Na leaf	K/Na stem	K/Na root	Ca/Na leaf	Ca/Na stem	Ca/Na root
Control (0 g L^−1^)	0.73 ± 0.005 i	1.51 ± 0.17 c	7.27 ± 0.02 b	0.45 ± 0.03 g	0.95 ± 0.02 c	7.12 ± 1.15 a
15 g L^−1^	0.57 ± 0.009 l	1.01 ± 0.12 de	1.74 ± 0.02 f	0.52 ± 0.009 de	0.70 ± 0.13 d	2.53 ± 0.13 c
30 g L^−1^	0.59 ± 0.01 l	1.15 ± 0.10 d	0.82 ± 0.01 i	0.49 ± 0.01 ef	0.79 ± 0.15 d	1.85 ± 0.23 d
50 g L^−1^	0.98 ± 0.01 g	1.56 ± 0.03 c	1.94 ± 0.05 e	0.54 ± 0.001 d	0.70 ± 0.003 d	1.49 ± 0.02 d
100 g L^−1^	0.11 ± 0.001 n	0.13 ± 0.005 o	0.11 ± 0.002 q	0.67 ± 0.001 c	0.36 ± 0.01 f	0.30 ± 0.01 ef
	*Juncus maritimus*					
NaCl	K/Na shoot	K/Na rhizome	K/Na root	Ca/Na shoot	Ca/Na rhizome	Ca/Na root
Control (0 g L^−1^)	3.63 ± 0.03 b	1.41 ± 0.18 c	2.16 ± 0.22 d	0.93 ± 0.03 b	1.56 ± 0.03 b	7.49 ± 0.65 a
15 g L^−1^	1.56 ± 0.01 e	0.97 ± 0.01 de	0.29 ± 0.01 no	0.44 ± 0.01 g	0.51 ± 0.02 e	1.85 ± 0.01 d
30 g L^−1^	1.44 ± 0.07 f	0.47 ± 0.003 i–m	0.37 ± 0.03 mn	0.31 ± 0.01 i	0.27 ± 0.01 f	0.68 ± 0.03 e
50 g L^−1^	0.47 ± 0.001 m	0.40 ± 0.01 l–n	0.13 ± 0.005 pq	0.11 ± 0.004 n	0.12 ± 0.002 h	0.20 ± 0.04 ef
100 g L^−1^	1.75 ± 0.005 d	0.29 ± 0.01 m–o	0.10 ± 0.01 q	0.48 ± 0.01 fg	0.11 ± 0.001 h	0.19 ± 0.001 ef
	*Phragmites australis*					
NaCl	K/Na shoot	K/Na rhizome	K/Na root	Ca/Na shoot	Ca/Na rhizome	Ca/Na root
Control (0 g L^−1^)	8.82 ± 0.13 a	3.03 ± 0.27 b	8.37 ± 0.001 a	1.35 ± 0.04 a	3.36 ± 0.20 a	3.84 ± 0.001 b
15 g L^−1^	3.23 ± 0.07 c	0.73 ± 0.05 fg	0.54 ± 0.001 l	0.53 ± 0.01 d	0.37 ± 0.01 f	0.53 ± 0.001 ef
30 g L^−1^	1.73 ± 0.001 d	0.62 ± 0.02 g–i	0.18 ± 0.001 o–q	0.29 ± 0.001 i	0.30 ± 0.02 f	0.38 ± 0.001 ef
50 g L^−1^	1.57 ± 0.03 e	0.24 ± 0.01 no	0.10 ± 0.001 q	0.38 ± 0.03 h	0.13 ± 0.01 gh	0.22 ± 0.001 ef
100 g L^−1^	0.11 ± 0.001 n	0.54 ± 0.01 h–l	0.08 ± 0.001 q	0.10 ± 0.001 n	0.11 ± 0.004 h	0.04 ± 0.001 f
Statistics	Two-way ANOVA—F (*P*)					
NaCl	5193.7 (<0.001)	621.9 (<0.001)	9969.6 (<0.001)	724.4 (<0.001)	667.2 (<0.001)	350.1 (<0.001)
species	8207.1 (<0.001)	52.2 (<0.001)	1803.4 (<0.001)	417.3 (<0.001)	108.0 (<0.001)	96.4 (<0.001)
NaCl x species	3846.9 (<0.0001)	86.3 (<0.001)	1309.1 (<0.001)	481.5 (<0.001)	163.7 (<0.001)	25.4 (<0.001)

**Table 4 plants-12-01737-t004:** Bioaccumulation and translocation factors (BAF, TF) of sodium (Na) in the four halophyte plant species treated with 0, 15, 30, 50, and 100 g L^−1^ NaCl (corresponding to 0, 257, 514, 856, and 1712 mM NaCl, respectively) for 28 days. Values are the mean ± SD of two or three extractions (*n* = 2–3) derived from bulked vegetal material of four plants for each species and NaCl treatment. Results of the two-way ANOVA (*p* ≤ 0.05) for the effects of NaCl treatments and species and of their interaction are shown. When the interaction between factors was significant, the Fisher LSD-test (*p* ≤ 0.05) was applied: significantly different data are followed by different letters in the same column.

	*Suaeda fruticosa*			
NaCl	BAF_shoot_	BAF_root_	BAF_whole plant_	TF
Control (0 g L^−1^)	–	–	–	11.06 ± 0.39 c
15 g L^−1^	8.31 ± 0.02 a	0.87 ± 0.07 cd	9.18 ± 0.56 a	9.55 ± 0.24 cd
30 g L^−1^	2.27 ± 0.11 c	0.26 ± 0.07 fg	2.06 ± 0.04 cd	9.09 ± 2.75 cd
50 g L^−1^	1.61 ± 0.26 d	0.32 ± 0.07 fg	1.90 ± 0.33 e	5.04 ± 0.37 e
100 g L^−1^	0.55 ± 0.03 f	0.33 ± 0.07 fg	1.05 ± 0.05 fg	1.70 ± 0.44 fg
	*Halocnemum strobilaceum*			
NaCl	BAF_shoot_	BAF_root_	BAF_whole plant_	TF
Control (0 g L^−1^)	–	–	–	35.43 ± 1.75 a
15 g L^−1^	4.32 ± 0.31 b	0.28 ± 0.08 fg	4.60 ± 0.40 b	16.23 ± 3.73 b
30 g L^−1^	2.32 ± 0.11 c	0.29 ± 0.05 fg	2.61 ± 0.06 c	8.07 ± 1.75 d
50 g L^−1^	1.21 ± 0.04 e	0.11 ± 0.02 g	1.32 ± 0.06 f	11.59 ± 2.22 c
100 g L^−1^	0.46 ± 0.10 f	0.14 ± 0.07 g	0.61 ± 0.17 g	3.38 ± 0.87 ef
	*Juncus maritimus*			
NaCl	BAF_shoot_	BAF_underground_	BAF_whole plant_	TF
Control (0 g L^−1^)	–	–	–	0.80 ± 0.10 fg
15 g L^−1^	1.01 ± 0.07 e	1.63 ± 0.23 b	2.64 ± 0.29 c	0.62 ± 0.05 g
30 g L^−1^	0.43 ± 0.02 fg	1.64 ± 0.21 b	2.06 ± 0.24 de	0.26 ± 0.02 g
50 g L^−1^	0.47 ± 0.06 f	1.43 ± 0.22 b	1.90 ± 0.27 e	0.33 ± 0.01 g
100 g L^−1^	0.06 ± 0.01 h	0.99 ± 0.17 c	1.05 ± 0.18 fg	0.07 ± 0.003 g
	*Phragmites australis*			
NaCl	BAF_shoot_	BAF_underground_	BAF_whole plant_	TF
Control (0 g L^−1^)	–	–	–	0.42 ± 0.11 g
15 g L^−1^	0.31 ± 0.02 f–h	2.00 ± 0.19 a	2.31 ± 0.21 c–e	0.15 ± 0.01 g
30 g L^−1^	0.08 ± 0.05 gh	0.53 ± 0.10 ef	0.61 ± 0.15 g	0.15 ± 0.07 g
50 g L^−1^	0.05 ± 0.01 h	0.87 ± 0.04 cd	0.92 ± 0.05 fg	0.06 ± 0.01 g
100 g L^−1^	0.08 ± 0.04 gh	0.68 ± 0.12 de	0.76 ± 0.16 g	0.11 ± 0.03 g
Statistics	Two-way ANOVA—F (*P*)			
NaCl	557.3 (<0.001)	39.48 (<0.001)	378.39 (<0.001)	74.03 (<0.001)
species	572.2 (<0.001)	142.69 (<0.001)	142.60 (<0.001)	287.96 (<0.001)
NaCl x species	171.8 (<0.0001)	11.63 (<0.001)	75.39 (<0.001)	41.51 (<0.001)

## Data Availability

The data presented in this study are available on request from the corresponding authors. The data are not publicly available due to the restriction policy of the coauthors’ affiliations.

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
