# Peer review of "Response to Hypersalinity of Four Halophytes Growing in Hydroponic Floating Systems: Prospects in the Phytomanagement of High Saline Wastewaters and Extreme Environments"

_plants, 2023, doi:10.3390/plants12091737_

Round 1
Reviewer 1 Report
Overall, this is a well-written and informative paper. However, there are some areas that need to be improved to make it more concise and clear. Here are my suggestions:
- The abstract is lengthy and can be refined to better highlight the paper's main findings. Shorten the introduction and focus on the four halophytes studied and their mechanisms of salt tolerance. Also, describe the most significant results and their implications for phytomanagement of hypersaline environments and high saline wastewaters.
- The introduction is currently only two paragraphs. Consider breaking down the different paragraphs to better organize the content.
- The results section should be condensed to present only the key findings and most important results. Avoid presenting too much data that may be confusing or overwhelming for the readers.
- In the figures, only compare between groups instead of comparing everything together, to make the comparisons clearer and easier to understand.
- In the tables, only compare within rows or columns instead of comparing everything together, to make the comparisons clearer and easier to understand.
- The discussion section can be further improved by highlighting the most significant and novel findings of the study, and their implications for phytomanagement of hypersaline environments and high saline wastewaters.
- The conclusion section should be brief but clear, highlighting the main findings and their implications for future research and practical applications.
- The references section needs to be more focused, with only 45-60 references that are directly relevant to the study. Consider removing irrelevant or outdated references.
Overall, with these revisions, the paper will be more focused and easier to read, with clear implications for phytomanagement of hypersaline environments and high saline wastewaters.
Reviewer 2 Report
The idea behind this research is excellent and there is very little similar data in the literature. However, you used aerated hydroponics rather than an ebb and flow system of culture. You also used NaCl to salinise the culture medium, ignoring previous work that has demonstrated the importance of maintaining the Ca/Na ratio, at least for some species. The method you used would likely not have been questioned twenty years ago, but now the method of choice would be an ebb and flow system using an inert medium like sand to support the plants and salinisation with artificial seawater. You need to justify using the method you used and comment in the Discussion.
Given the method you have used, you should provide some measures of oxygen concentrations in the medium and reassure the reader that the responses were not a consequence of low oxygen concentration or due to physical damage of fine roots as has been reported for Suaeda maritima.
You used high concentrations of NaCl and should at least look at the effect of these on the activities of other ions in your culture medium, using a program such as MINTEQ AQ. You need to be able to re-assure the reader that the changes you describe were not, for example, due to a micronutrient deficiency.
You included fresh weight data, which should, along with water content data, be removed. It is not possible to blot succulent leaves and especially roots uniformly. Fresh weight measurements for shoots are best made on unwashed material pre-dawn.
The presentation is dense and wordy, perhaps because there is so much data. I believe your paper would have more impact were you to draw out the most important findings and write about those. Put the rest of the data in the Supplementary Material. There are many areas of the text where division into paragraphs would help the reader with one topic per paragraph (some places where the text should be divided are shown in my marked ms). I also think the abstract is too long: much of the opening seven lines could, for example, be deleted.
The use of English is generally good, but there are places where the text needs revision. I have highlighted some of these in a copy of your manuscript.
Concentrations should be given in mM values throughout; concentrations expressed in this way are indicative of osmotic potentials across elements.
There is more data on the species used in eHALOPH and the following papers might be useful
Pujol JA, Calvo JF, Ramirez-Diaz L (2001) Seed germination, growth, and osmotic adjustment in response to NaCl in a rare succulent halophyte from southeastern Spain. Wetlands 21 (2):256-264
Zorb C, Sumer A, Sungur A, Flowers TJ, Ozcan H (2013) Ranking of 11 coastal halophytes from salt marshes in northwest Turkey according their salt tolerance. Turkish Journal of Botany 37 (6):1125-1133. doi:10.3906/bot-1205-29
